# Explaining the Efficacy of Counterfactually Augmented Data

**Divyansh Kaushik, Amrith Setlur, Eduard Hovy, Zachary C. Lipton**
Carnegie Mellon University
Pittsburgh, PA, USA
`{dkaushik, asetlur, hovy, zlipton}@cmu.edu`

## Abstract

In attempts to produce machine learning models less reliant on spurious patterns in NLP datasets, researchers have recently proposed curating counterfactually augmented data (CAD) via a human-in-the-loop process in which given some documents and their (initial) labels, humans must revise the text to make a counterfactual label applicable. Importantly, edits that are not necessary to flip the applicable label are prohibited. Models trained on the augmented (original *and* revised) data appear, empirically, to rely less on semantically irrelevant words and to generalize better out of domain. While this work draws loosely on causal thinking, the underlying causal model (even at an abstract level) and the principles underlying the observed out-of-domain improvements remain unclear. In this paper, we introduce a toy analog based on linear Gaussian models, observing interesting relationships between causal models, measurement noise, out-of-domain generalization, and reliance on spurious signals. Our analysis provides some insights that help to explain the efficacy of CAD. Moreover, we develop the hypothesis that while adding noise to causal features should degrade both in-domain and out-of-domain performance, adding noise to non-causal features should lead to relative improvements in out-of-domain performance. This idea inspires a speculative test for determining whether a feature attribution technique has identified the *causal spans*. If adding noise (e.g., by random word flips) to the highlighted spans degrades both in-domain and out-of-domain performance on a battery of challenge datasets, but adding noise to the complement gives improvements out-of-domain, this suggests we have identified causal spans. Thus, we present a large-scale empirical study comparing spans edited to create CAD to those selected by attention and saliency maps. Across numerous challenge domains and models, we find that the hypothesized phenomenon is pronounced for CAD.

## 1 Introduction

Despite machine learning (ML)'s many practical breakthroughs, formidable obstacles obstruct its deployment in consequential applications. Of particular concern, these models have been shown to rely on spurious signals, such as surface-level textures in images (Jo & Bengio, 2017; Geirhos et al., 2018), and background scenery—even when the task is to recognize foreground objects (Beery et al., 2018). Other studies have uncovered a worrisome reliance on gender in models trained for the purpose of recommending jobs (Dastin, 2018), and on race in prioritizing patients for medical care (Obermeyer et al., 2019). Moreover, while modern ML performs remarkably well on independent and identically distributed (iid) holdout data, performance often decays catastrophically under both naturally occurring and adversarial distribution shift (Quionero-Candela et al., 2009; Sugiyama & Kawanabe, 2012; Szegedy et al., 2014; Ovadia et al., 2019; Filos et al., 2020).

These two problems: (i) reliance on semantically irrelevant signals, raising concerns about bias; and (ii) the brittleness of models under distributions shift; might appear unrelated, but share important conceptual features. Concerns about bias stem in part from principles of procedural fairness (Blader & Tyler, 2003; Miller, 2017; Grgic-Hlaca et al., 2018; Lipton et al., 2018), according to which decisions should be based on qualifications, not on distant proxies that are spuriously associated with the outcome of interest. Arguably one key distinction of an actual qualification might be that it

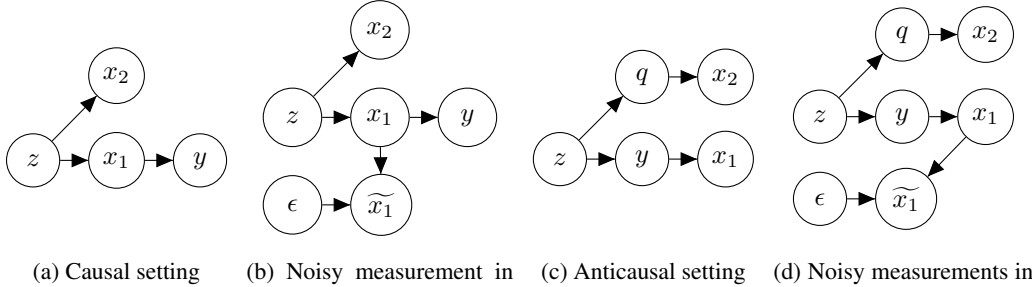

(a) Causal setting    (b) Noisy measurement in causal setting    (c) Anticausal setting    (d) Noisy measurements in anticausal setting

Figure 1: Toy causal models with one hidden confounder. In 1a and 1c, the observed covariates are $x_1, x_2$. In 1b and 1d, the observed covariates are $\widetilde{x_1}, x_2$. In all cases, $y$ denotes the label.

actually exerts causal influence on the outcome of interest. In an interesting parallel, one line of work on distribution shift has focused on causal graphical models, addressing settings where some parts of the model remain stable over time but others do not. One common assumption is that the relationship between the target and its direct causal ancestors remains invariant (Peters et al., 2016; Ghassami et al., 2017; Rojas-Carulla et al., 2018; Kuang et al., 2018; Magliacane et al., 2018; Christiansen & Peters, 2020; Weichwald & Peters, 2020). While these papers contribute insight, they focus on toy settings, with few variables related by a known model. However, in complex domains with high-dimensional data, what variables are relevant and what graph relates them is typically unclear.

Recently in NLP, Kaushik et al. (2020) proposed *Counterfactually Augmented Data (CAD)*, injecting causal thinking into real world settings by leveraging human-in-the-loop feedback to identify causally relevant features (versus those that merely happen to be predictive due to confounding). Human editors are presented with document-label pairs and tasked with editing documents to render counterfactual labels applicable. The instructions restrict editors to only make modifications that are necessary to flip the label's applicability. The key result is that many spurious correlations present in the original dataset are absent in the CAD. In case of sentiment analysis, Kaushik et al. (2020) demonstrated that linear classifiers trained to predict the sentiment of movie reviews based on bag-of-words representations assign high-magnitude weights to seemingly irrelevant terms, including "will", "my", "has", "especially", and "script", among others. Notably, "horror" featured among the most negative terms, while "romance" featured among the most positive, despite both communicating genre, not sentiment. Interestingly, in the revised data, each "horror" review retains the word "horror" (per the instruction not to make unnecessary edits) but is associated with the opposite sentiment label. Models trained on the augmented data (original *and* revised) perform well on both original and revised data, and assign little weight to the associated but irrelevant terms. Intuitively, one might imagine that the spurious patterns would generalize less reliably out of domain. Most consumer products do not belong to movie genres, but words like "excellent" and "awful" continue to connote positive and negative sentiment, respectively. Indeed, Kaushik et al. (2020) demonstrated that models trained on CAD enjoyed out-of-domain performance benefits on Tweets, and Amazon and Yelp reviews.

In this paper, we make some initial attempts towards explaining CAD's efficacy. While CAD plainly draws on causal thinking, (invoking interventions and counterfactuals), foundational questions remain open: What is the assumed causal structure underlying settings where CAD might be effective? What are the principles underlying its out-of-domain benefits? Must humans *really* intervene, or could automatic feature attribution methods, e.g., attention (DeYoung et al., 2020), or cheaper feedback mechanisms, e.g., feature feedback (Zaidan et al., 2007), produce similar results?

To begin, we consider linear Gaussian models (Figure 1; Wright, 1934), with the following goals: to (i) gain qualitative insights into when a predictor might rely on spurious signals in the first place; and (ii) provide a mechanism of action to explain the efficacy of CAD. First, we analyze the causal setting (features cause the label). When the features share a common cause and a predictor is well-specified (linear), it will assign zero weight (in expectation) to non-causal features. However, when the causal features are subject to observation noise (measurement error), the non-causal features are assigned non-zero weight. Conversely, when we inject noise on non-causal features, predictors rely more on causal features, which we expect to result in better out-of-domain generalization. In the

causal framework, we observe that CAD might be usefully formalized as a process analogous to intervening on the causal features, thus d-separating the label from the non-causal features (Pearl, 1985). Alternatively, we might conceptualize CAD with an anticausal model (Schölkopf et al., 2012). In this setup, the label of interest is one of several latent attributes that directly causes some (but not all features). In this interpretation, we imagine that we have intervened on the label and the editor's role is to simulate the counterfactual document that would flow from the alternative label, holding other attributes constant. Note that this too d-separates the label from the spurious correlate. In both cases, any model trained on the resulting data ought to rely only on the causal features.

Our toy abstraction points to a useful diagnostic test. If indeed CAD involves interventions on spans that are (in some sense) analogous to the causal features in our toy model, then injecting noise on these words should increase model reliance on the non-causal features and thus (in general) lead to deteriorating performance out-of-domain. On the other hand, injecting noise on the non-causal features should lead the model to rely more on the causal features, leading to improved performance out of domain. Through a series of large-scale empirical experiments addressing sentiment analysis and natural language inference (NLI) tasks, we inject noise on the spans marked as causal vs non-causal. We compare the effects of injecting noise on the spans revised by the CAD editors, the spans selected through feature feedback (Zaidan et al., 2007), and to spans selected automatically using feature attribution heuristics such as attention- and gradient-based saliency methods. If indeed the hypotheses that (i) identifying causal features requires human intervention; and (ii) models relying on causal features generalize better out of domain; hold, we might expect that (compared to automatic attribution methods) noising human-provided rationales would deteriorate out-of-domain performance, while noising non-rationales should prove beneficial.

We show that an SVM sentiment analysis model trained on the original $1.7k$ IMDb reviews from Kaushik et al. (2020) obtains $87.8\%$ accuracy on the IMDb test set and $79.9\%$ on Yelp reviews but when all *rationales* are replaced with noise, the classifier experiences $\approx 11\%$ drop on in-sample accuracy and an even bigger drop of $\approx 28.7\%$ on Yelp. However, as *non-rationales* are replaced with noise, in-domain accuracy goes down by $\approx 10\%$ but out-of-domain accuracy increases by $1.5\%$. Similarly, in NLI, the accuracy of a BERT classifier fine-tuned on a subsample of e-SNLI (DeYoung et al., 2020) goes down by $\approx 20\%$ when *rationales* are replaced with noise, whereas the out-of-domain accuracy goes down by $21.3$–$31.5\%$ on various datasets. If *non-rationales* are replaced with noise, in-sample accuracy goes down by $6.2\%$ but out of domain accuracy drops by only $2.3$–$5.5\%$. Similar patterns are observed across both tasks, on all datasets and models. However, when using attention masks, the resulting changes in model performance do not appear to follow these trends. In another test to probe whether human feedback is indeed necessary to produce datasets with the observed quantitative results of CAD, we experiment with style transfer methods for converting *Positive* reviews into *Negative* and vice versa. Compared to an SVM classifier trained on style-transfer-augmented data, training on CAD leads to a gain of $5$–$16.4\%$ in accuracy on Amazon and $3.7$–$17.8\%$ on Yelp. Similarly, a BERT classifier fine-tuned on CAD outperforms the same classifier fine-tuned on style-transfer-augmented data by $4.9$–$21.5\%$ on Amazon and $1.9$–$9.5\%$ on Yelp.

## 2 RELATED WORK

NLP papers on spurious associations have addressed social biases (Dixon et al., 2018; Zhao et al., 2018; Kiritchenko & Mohammad, 2018; Dinan et al., 2019; May et al., 2019), spurious signals owing to annotation heuristics (Gururangan et al., 2018; Poliak et al., 2018), and artifacts from automatic data generation (Chen et al., 2016; Kaushik & Lipton, 2018), Researchers have also demonstrated vulnerabilities to synthetic transformations, such as distractor phrases (Jia & Liang, 2017; Wallace et al., 2019), document paraphrases (Iyyer et al., 2018; Pfeiffer et al., 2019), and synthetic but meaning-preserving modifications (Ribeiro et al., 2018; Glockner et al., 2018; Shen et al., 2018).

Researchers have proposed incorporating human feedback solicited through a variety of mechanisms including highlighting *rationales*, spans of text indicative of the label (Zaidan et al., 2007; Zaidan & Eisner, 2008; Poulis & Dasgupta, 2017). To combat gender stereotypes, Lu et al. (2018); Zmigrod et al. (2019); Maudslay et al. (2019) describe data augmentation approaches that programmatically alter text. More recently, Kaushik et al. (2020) employed crowd workers to edit text to make an opposite label applicable. Through their experiments they show that classifiers trained on CAD generalize well out of domain. Teney et al. (2020) show the benefits of CAD in computer vision and

NLP, and Srivastava et al. (2020) employ crowdworkers to augment their training data to capture potential unmeasured variables. A growing body of work has also looked at reducing reliance on spurious correlations by exploiting the stability of relationships between the target variable and its (graph) neighbors. Peters et al. (2016) propose *invariant causal prediction* to obtain a causal predictor from multiple datasets. Ghassami et al. (2017) discuss a similar approach but do not assume that the exogenous noise of the target variable stays fixed among environments. They also demonstrate the benefits of their approach (compared to Peters et al. (2016)) in identifying all direct ancestors of the target variable. Arjovsky et al. (2019) propose *invariant risk minimization*, with the goal of learning a data representation such that the optimal predictor is shared across environments.

## 3 ANALYSIS OF A TOY MODEL

We briefly review the OLS estimator for the model $Y = X\beta + \epsilon$, where $Y \in \mathrm{R}^n$ is the target, $X \in \mathrm{R}^{n \times p}$ the design matrix, $\beta \in \mathrm{R}^p$ the coefficient vector we want to estimate, and $\epsilon \sim \mathcal{N}(0, \sigma_\epsilon^2 \mathbf{I}_n)$ an iid noise term. The OLS estimate $\beta^{ols}$ is given by $\mathrm{Cov}(X, X)\beta^{ols} = \mathrm{Cov}(X, Y)$. Representing $\mathrm{Var}[X_i]$ as $\sigma_{x_i}^2$ and $\mathrm{Cov}(X_i, X_j)$ as $\sigma_{x_i, x_j}$, if we observe only two covariates ($p = 2$), then:

$$\beta_1^{ols} = \frac{\sigma_{x_2}^2 \sigma_{x_1, y} - \sigma_{x_1, x_2} \sigma_{x_2, y}}{\sigma_{x_1}^2 \sigma_{x_2}^2 - \sigma_{x_1, x_2}^2} \qquad \beta_2^{ols} = \frac{\sigma_{x_1}^2 \sigma_{x_2, y} - \sigma_{x_1, x_2} \sigma_{x_1, y}}{\sigma_{x_1}^2 \sigma_{x_2}^2 - \sigma_{x_1, x_2}^2} \tag{1}$$

Our analysis adopts the structural causal model (SCM) framework (Pearl, 2009), formalizing causal relationships via Directed Acyclic Graphs (DAGs). Each edge of the form $A \to B \in \mathcal{E}$ in a DAG $\mathcal{G} = (\mathcal{V}, \mathcal{E})$ indicates that the variable $A$ is (potentially) a direct cause of variable $B$. All measured variables $X \in \mathcal{V}$ in the model are deterministic functions of their corresponding parents $\mathrm{Pa}(X) \subseteq \mathcal{V}$ and a set of jointly independent noise terms. For simplicity, we work with linear Gaussian SCMs in the presence of a single confounder where each variable is a linear function of its parents and the noise terms are assumed to be additive and Gaussian. We look at both causal and anticausal learning settings. In the former, we assume that a document causes the applicability of the label (as in annotation, where the document truly causes the label). In the latter interpretation, we assume that the label is one latent variable (among many) that causes features of the document (as when a reviewer's "actual sentiment" influences what they write). For simplicity, we assume that the latent variables are correlated due to confounding but that each latent causes a distinct set of observed features. Without loss of generality, we assume that all variables have zero mean. Both DAGs contain the four random variables $z, x_1, x_2, y$ and the anticausal DAG also contains some additional latent variables $q$ (Figure 1). The derivations are standard and are included in Appendix A.

### 3.1 THE CAUSAL SETTING

We now focus on the causal setting (Figure 1a, 1b) Let the Gaussian SCM be defined as follows where the noise term for variable $x$ is defined as $u_x$:

$$\begin{aligned}
z &= u_z, & u_z &\sim \mathcal{N}(0, \sigma_{u_z}^2) \\
x_1 &= bz + u_{x_1}, & u_{x_1} &\sim \mathcal{N}(0, \sigma_{u_{x_1}}^2) \\
x_2 &= cz + u_{x_2}, & u_{x_2} &\sim \mathcal{N}(0, \sigma_{u_{x_2}}^2) \\
y &= ax_1 + u_y, & u_y &\sim \mathcal{N}(0, \sigma_{u_y}^2).
\end{aligned} \tag{2}$$

Applying OLS, we obtain $\beta_1^{ols} = a$ and $\beta_2^{ols} = 0$. However, consider what happens if we only observe $x_1$ via a noisy proxy $\widetilde{x_1} \sim \mathcal{N}(x_1, \sigma_{u_{x_1}}^2 + \sigma_{\epsilon_{x_1}}^2)$ (Figure 1b). Assuming, $\epsilon_{x_1} \perp\!\!\!\perp (x_1, x_2, y)$, from Eq. 1 we get the estimates $\widehat{\beta_1^{ols}}$ and $\widehat{\beta_2^{ols}}$ (Eq. 3) in the presence of observation noise on $x_1$.

$$\begin{aligned}
\widehat{\beta_1^{ols}} &= \frac{a(\sigma_{u_z}^2(b^2 \sigma_{u_{x_2}}^2 + c^2 \sigma_{u_{x_1}}^2) + \sigma_{u_{x_1}}^2 \sigma_{u_{x_2}}^2)}{\sigma_{u_z}^2(b^2 \sigma_{u_{x_2}}^2 + c^2 \sigma_{u_{x_1}}^2) + \sigma_{u_{x_1}}^2 \sigma_{u_{x_2}}^2 + \sigma_{\epsilon_{x_1}}^2(c^2 \sigma_{u_z}^2 + \sigma_{u_{x_2}}^2)} \\
\widehat{\beta_2^{ols}} &= \frac{acb\sigma_{\epsilon_{x_1}}^2 \sigma_{u_z}^2}{\sigma_{u_z}^2(b^2 \sigma_{u_{x_2}}^2 + c^2 \sigma_{u_{x_1}}^2) + \sigma_{u_{x_1}}^2 \sigma_{u_{x_2}}^2 + \sigma_{\epsilon_{x_1}}^2(c^2 \sigma_{u_z}^2 + \sigma_{u_{x_2}}^2)}
\end{aligned} \tag{3}$$

As we can see, $\widehat{\beta_1^{ols}} \propto \frac{1}{\sigma_{\epsilon_{x_1}}^2}$. This shows us that as $\sigma_{\epsilon_{x_1}}^2$ increases, $|\widehat{\beta_1^{ols}}|$ (the magnitude of the coefficient for $x_1$) decreases and $|\widehat{\beta_2^{ols}}|$ (the magnitude of the coefficient for $x_2$) increases. The

asymptotic OLS estimates in the presence of infinite observational noise is $\lim_{\sigma^2_{\epsilon_{x1}} \to \infty} \widehat{\beta_1^{ols}} = 0$, whereas $\widehat{\beta_2^{ols}}$ converges to a finite non-zero value. On the other hand, observing a noisy version of $x_2$ will not affect our OLS estimates if there is no measurement error on $x_1$.

These simple graphs provide qualitative insights into when we should expect a model to rely on spurious patterns. In the causal setting, under perfect measurement, the causal variable d-separates the non-causal variable from the label (Figure 1a). However, under observation noise, a predictor will rely on the non-causal variable (Eq. 3). Moreover, when the causal feature is noisily observed, additional observation noise on non-causal features yields models that are more reliant on causal features. We argue that while review text is not noisily observed per se, learning with imperfect feature representations acquired by training deep networks on finite samples has an effect that is analogous to learning with observation noise.

**Connection to Counterfactually Augmented Data**  In the causal setting, intervening on the causal feature, d-separates the label $y$ from the non-causal feature $x_2$, and thus models trained on samples from the interventional distribution will rely solely on the causal feature, even when it is noisily observed. We argue that in a qualitative sense, the process of generating CAD resembles such an intervention, however instead of intervening randomly, we ensure that for each example, we produce two sets of values of $x_1$, one such that the label is applicable and one such that it is not applicable. One is given in the dataset, and the other is produced via the revision.

## 3.2 An Anticausal Interpretation

Alternatively, rather than thinking of features causing the applicable label, we might think of the "causal feature" as a direct effect of the label (not a cause). In this case, so long as the relationship is truly not deterministic, even absent noisy observation, conditioning on the causal feature does not d-separate the label from the non-causal feature and thus models should be expected to assign weight to both causal and non-causal variables.

As in the causal setting, as we increase observation noise on the causal variable, the weight assigned to the non-causal variable should increase. Conversely, as in the causal setting with observation noise on $x_1$, as observation noise on the non-causal feature $x_2$ increases, we expect the learned predictor to rely more on the causal feature. We derive the OLS coefficients (including under the presence of observational noise, Fig. 1d) in this setting in Appendix A.2.

**Connection to Counterfactually Augmented Data**  In this interpretation, we think of CAD as a process by which we (the designers of the experiment) intervene on the label itself and the human editors, play the role of a simulator that we imagine to be capable of generating a counterfactual example, holding all other latent variables constant. In the sentiment case, we could think of the editors as providing us with the review that would have existed had the sentiment been flipped, holding all other aspects of the review constant. Note that by intervening on the label, we d-separate it from the spurious correlate $x_2$ (Figure 1c).

## 3.3 Insights and Testable Hypotheses

In both the causal and anticausal models, the mechanism underlying the causal relationship that binds $x_1$ to $y$ (regardless of direction) is that binding language to a semantic concept (such as sentiment), which we expect to be more stable across settings than the more capricious relationships among the background variables, e.g., those linking genre and production quality.

In that spirit, if spans edited to generate *counterfactually revised data* (CRD) are analogous to the causal (or anticausal) variables, in the causal (or anticausal) graphs, then we might expect that noising those spans (e.g. by random word replacement) should lead to models that rely more on non-causal features and perform worse on out of domain data. On the other hand, we expect that noising unedited spans should have the opposite behavior, leading to degraded in-domain performance, but comparatively better out-of-domain performance. In the remainder of the paper, we investigate these hypotheses, finding evidence that qualitatively confirms the predictions of our theory.

We freely acknowledge the speculative nature of this analysis and concede that the mapping between the messy unstructured data we wish to model and the neatly disentangled portrait captured by our linear Gaussian models leaves a gap to be closed through further iterations of theoretical refinement and scientific experiment. Ultimately, our argument is not that this simple analysis fully accounts for counterfactually augmented data but instead that it is a useful abstraction for formalizing two (very different) perspectives on how to conceive of CAD, and for suggesting interesting hypotheses amenable to empirical verification.

## 4 EMPIRICAL RESULTS

If spans marked as rationales by humans via editing or highlighting are analogous to causal features, then noising those spans should lead to models that rely more on non-causal features and thus perform worse on out-of-domain data, and noising the unmarked spans (analogous to non-causal features) should have the opposite behavior. In this section, we test these hypotheses empirically on real-world datasets. Additionally, we investigate whether the feedback from human workers is yielding anything qualitatively different from what might be seen with spans marked by automated feature attribution methods such as attention and saliency. Along similar, lines we ask whether CAD in the first place offers qualitative advantages over what might be achieved via automatic sentiment-flipping methods through experiments with text style transfer algorithms.

We conduct experiments on sentiment analysis (Zaidan et al., 2007; Kaushik et al., 2020) and NLI (DeYoung et al., 2020). All datasets are accompanied with human feedback (tokens deemed relevant to the label's applicability) which we refer to as *rationales*. For the first set of experiments, we rely on four models: Support Vector Machines (SVMs), Bidirectional Long Short-Term Memory Networks (BiLSTMs) with Self-Attention (Graves & Schmidhuber, 2005), BERT (Devlin et al., 2019), and Longformer (Beltagy et al., 2020). For the second set of experiments, we rely on four state-of-the-art style transfer models representative of different methodologies, each representative of a different approach to automatically generate new examples with flipped labels (Hu et al., 2017; Li et al., 2018; Sudhakar et al., 2019; Madaan et al., 2020). To evaluate classifier performance on the resulting augmented data, we consider SVMs, Naive Bayes (NB), BiLSTMs with Self Attention, and BERT. We relegate implementation details to Appendix B.

For sentiment analysis, we use SVM, BiLSTM with Self Attention, BERT, and Longformer models. In each document, we replace a fraction of *rationale* (or *non-rationale*) tokens with random tokens sampled from the vocabulary, and train our models, repeating the process 5 times. We perform similar experiments for NLI using BERT. As an individual premise-hypothesis pair is often not as long as a movie review, many pairs only have one or two words marked as *rationales*. To observe the effects from gradually injecting noise on *rationales* or *non-rationales*, we select only those premise-hypothesis pairs that have a minimum 10 tokens marked as *rationales*. Since no neutral pairs exist with 10 or more rationale tokens, we consider only a binary classification setting (entailment-contradiction), and downsample the majority class to ensure a 50:50 label split.

Figures 2 and 3 show the difference in mean accuracy over 5 runs. For all classifiers, as the noise in *rationales* increases, in-sample accuracy stays relatively stable compared to out-of-domain accuracy. An SVM classifier trained on the original $1.7k$ IMDb reviews from Kaushik et al. (2020) obtains $87.8\%$ accuracy on the IMDb test set and $79.9\%$ on Yelp reviews.[1] As a greater fraction of *rationales* are replaced with random words from the vocabulary, the classifier experiences a drop of $\approx 11\%$ by the time all *rationale* tokens are replaced with noise. However, it experiences an $28.7\%$ drop in accuracy on Yelp reviews. Similarly, on the same datasets, a fine-tuned BERT classifier sees its in-sample accuracy drop by $18.4\%$, and by $31.4\%$ on Yelp as *rationale* tokens replaced by noise go from 0 to $100\%$. However, as more *non-rationales* are replaced with noise, in-sample accuracy for SVM goes down by $\approx 10\%$ but increases by $1.5\%$ on Yelp. For BERT, in-sample accuracy decreases by only $16.1\%$ and only $13.6\%$ on Yelp (Also see Appendix Table 3, and Appendix Figure 4a).

We obtain similar results using *rationales* identified via feature feedback. An SVM classifier trained on reviews from Zaidan et al. (2007) sees in-sample accuracy drop by $11\%$, and accuracy on Yelp

---

[1]The out-of-domain evaluation sets in Kaushik et al. (2020) do not have 50:50 label split. We enforce this split to observe when a classifier approaches random baseline performance. All datasets can be found at https://github.com/acmi-lab/counterfactually-augmented-data

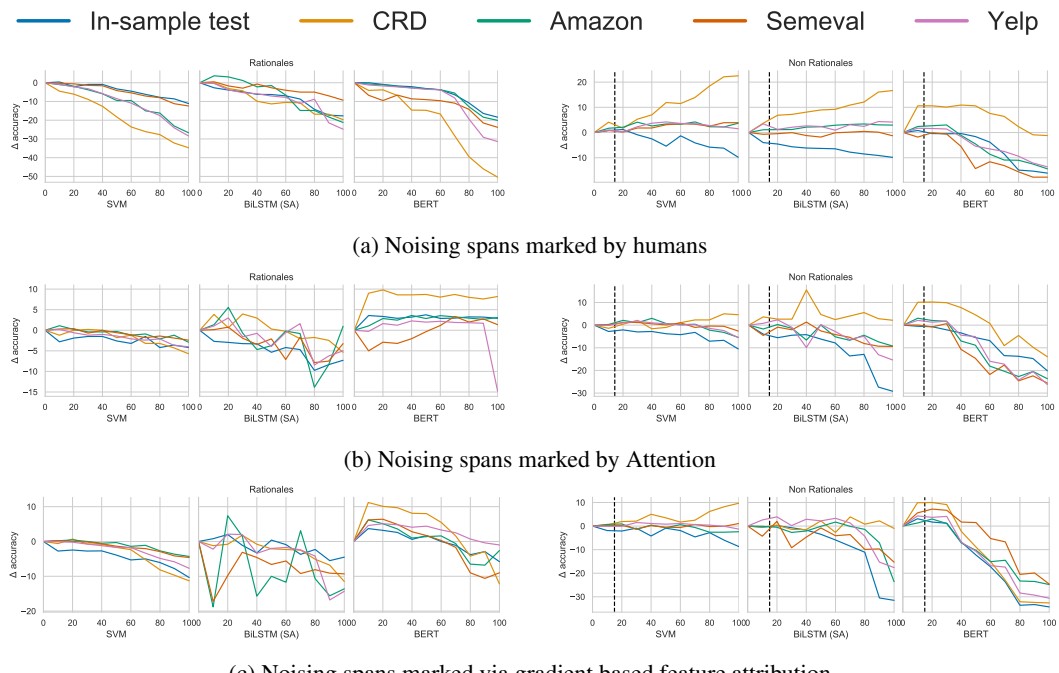

Figure 2: Change in classifier accuracy as noise is injected on *rationales/non-rationales* for IMDb reviews from Kaushik et al. (2020).

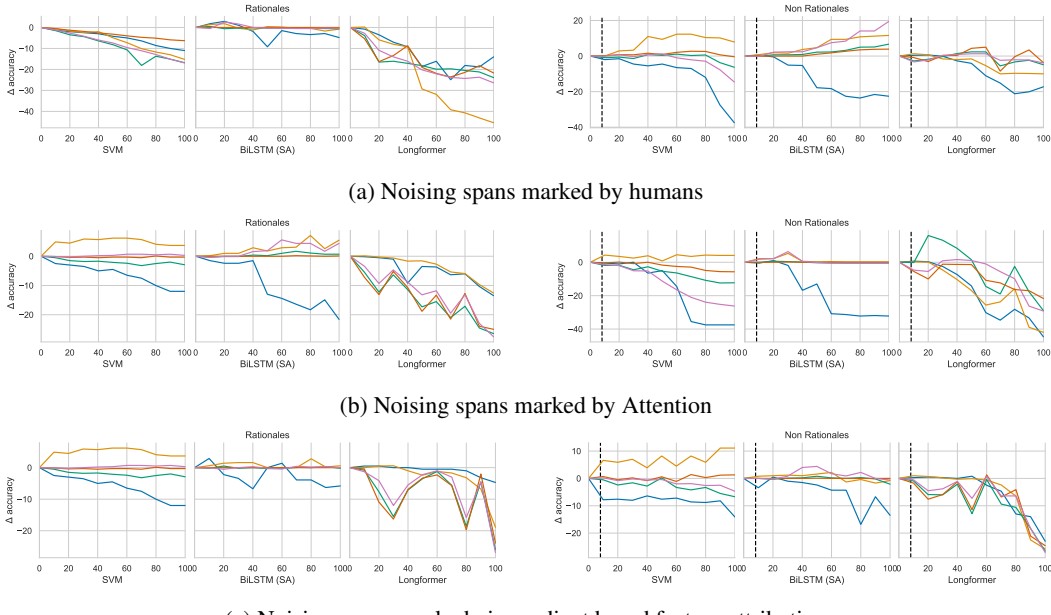

Figure 3: Change in classifier accuracy as noise is injected on *rationales/non-rationales* for IMDb reviews from Zaidan et al. (2007). In both Figures 2 and 3, the vertical dashed line indicates the fraction of median length of *non-rationales* equal to the median length of *rationales*.

drop by 16.9% as noise is inserted on *rationales* but goes down by 17.3% and 14.6%, respectively when noise is inserted in *non-rationales*. For Longformer, in-sample accuracy drops by 14% and accuracy on Yelp goes down by 26.4% compared to a drop of 17.3% and gain of 3.9%, respectively,

Table 1: Accuracy of BERT trained on SNLI (DeYoung et al., 2020) as noise is injected on human identified *rationales/non-rationales*. RP and RH are Revised Premise and Revised Hypothesis test sets in Kaushik et al. (2020). MNLI-M and MNLI-MM are MNLI (Williams et al., 2018) dev sets.

| Dataset | Percent noise added to train data rationales | | | | | | | | | | |
|---|---|---|---|---|---|---|---|---|---|---|---|
| | 0 | 10 | 20 | 30 | 40 | 50 | 60 | 70 | 80 | 90 | 100 |
| In-sample test | 91.6 | 90.7 | 90.0 | 88.9 | 87.3 | 86.2 | 84.4 | 80.2 | 78.0 | 72.2 | 71.9 |
| RP | 72.7 | 70.7 | 69.1 | 67.1 | 65.7 | 62.4 | 61.8 | 57.7 | 55.6 | 53.8 | 51.4 |
| RH | 84.7 | 80.8 | 80.4 | 79.5 | 77.2 | 75.7 | 73.3 | 67.7 | 64.0 | 57.9 | 53.2 |
| MNLI-M | 75.6 | 74.7 | 73.9 | 72.0 | 70.6 | 69.1 | 64.7 | 59.1 | 55.8 | 54.4 | 53.3 |
| MNLI-MM | 77.9 | 76.7 | 75.6 | 73.9 | 72.3 | 70.8 | 65.6 | 58.4 | 55.1 | 53.6 | 52.5 |
| Dataset | Percent noise added to train data non-rationales | | | | | | | | | | |
| | 0 | 10 | 20 | 30 | 40 | 50 | 60 | 70 | 80 | 90 | 100 |
| In-sample test | 91.6 | 91.4 | 91.3 | 90.9 | 90.8 | 89.9 | 89.0 | 88.7 | 87.8 | 86.7 | 85.4 |
| RP | 72.7 | 73.5 | 73.2 | 72.1 | 71.5 | 70.7 | 70.6 | 70.6 | 70.6 | 70.6 | 70.4 |
| RH | 84.7 | 83.6 | 82.6 | 81.9 | 81.3 | 81.1 | 80.5 | 79.8 | 79.4 | 79.4 | 79.2 |
| MNLI-M | 75.6 | 74.9 | 74.4 | 72.6 | 72.4 | 71.8 | 71.3 | 71.3 | 70.9 | 70.9 | 70.8 |
| MNLI-MM | 77.9 | 76.2 | 75.8 | 75.0 | 74.6 | 74.3 | 73.9 | 73.7 | 73.3 | 73.0 | 72.8 |

Table 2: Out-of-domain accuracy of models trained on original only, CAD, and original and *sentiment-flipped* reviews

| Training data | SVM | NB | BiLSTM (SA) | BERT |
|---|---|---|---|---|
| Accuracy on Amazon Reviews | | | | |
| CAD (3.4*k*) | **79.3** | **78.6** | **71.4** | **83.3** |
| Orig. & Hu et al. (2017) | 66.4 | 71.8 | 62.6 | 78.4 |
| Orig. & Li et al. (2018) | 62.9 | 65.4 | 57.6 | 61.8 |
| Orig. & Sudhakar et al. (2019) | 64.0 | 69.3 | 54.7 | 77.2 |
| Orig. & Madaan et al. (2020) | 74.3 | 73.0 | 63.8 | 71.3 |
| Orig. (3.4*k*) | 74.5 | 74.3 | 68.9 | 80.0 |
| Accuracy on Semeval 2017 (Twitter) | | | | |
| CAD (3.4*k*) | **66.8** | **72.4** | **58.2** | **82.8** |
| Orig. & Hu et al. (2017) | 60.9 | 63.4 | 56.6 | 79.2 |
| Orig. & Li et al. (2018) | 57.6 | 60.8 | 54.7 | 62.7 |
| Orig. & Sudhakar et al. (2019) | 59.4 | 62.6 | 54.9 | 72.5 |
| Orig. & Madaan et al. (2020) | 62.8 | 63.6 | 54.6 | 79.3 |
| Orig. (3.4*k*) | 63.1 | 63.7 | 50.7 | 72.6 |
| Accuracy on Yelp Reviews | | | | |
| CAD (3.4*k*) | **85.6** | **86.3** | **73.7** | **86.6** |
| Orig. & Hu et al. (2017) | 77.4 | 80.4 | 68.8 | 84.7 |
| Orig. & Li et al. (2018) | 67.8 | 73.6 | 63.1 | 77.1 |
| Orig. & Sudhakar et al. (2019) | 69.4 | 75.1 | 66.2 | 84.5 |
| Orig. & Madaan et al. (2020) | 81.3 | 82.1 | 68.6 | 78.8 |
| Orig. (3.4*k*) | 81.9 | 82.3 | 72.0 | 84.3 |

when noise is inserted in *non-rationales*. Similar patterns are observed across datasets and models (see Figure 3a, Appendix Table 6, and Appendix Figure 5a).[2]

---

[2]While similar trends are observed for both feature feedback and CAD, it is less clear how to incorporate feature feedback for training effectively with deep neural networks and pre-trained transformer architectures, whereas training (or fine-tuning) models on CAD is straightforward.

For NLI, the in-sample accuracy of BERT fine-tuned on an SNLI subsample drops by $\approx 20\%$ when *rationales* are replaced with noise, and out-of-domain accuracy goes down by 21.3–31.5% on various datasets (Table 10). Whereas, if *non-rationales* are replaced with noise, in-sample accuracy goes down by 6.2% but out-of-domain accuracy drops by only 2.3–5.5%. These results support our hypothesis that spans marked by humans as causing a label are analogous to causal variables.

Interestingly, in our NLI experiments, for various models the drops in both in-sample and out-of-domain accuracy are greater in magnitude when noise is injected in *rationales* versus when it is injected in *non-rationales*. This is opposite to what we observe in sentiment analysis. We conjecture that these results are due to the fact that in our experiment design for NLI, we only keep those premise-hypothesis pairs that contain at least 10 tokens marked as *rationales* so we can observe the difference in accuracy as the amount of noise increases. A consequence of this selection is that many pairs selected have many more tokens marked as *rationales* than *non-rationales*, whereas, in sentiment analysis this is the opposite. Hence, in NLI when some percentage of *rationales* are replaced by noise, this corresponds to many more edited tokens than when a corresponding percentage of *non-rationales* are noised.

To compare human feedback to automatic feature attribution methods such as attention (Bahdanau et al., 2015) and gradient based saliency methods (Li et al., 2016), we conduct the same set of experiments assuming tokens attended to (or not) by an attention based classifier (BiLSTM with Self-Attention) or identified as highly influential by a gradient based feature attribution method (salience scores) as new *rationales* (or *non-rationales*). In this case, unlike our findings with human feedback, we observe markedly different behavior than predicted by our analysis of the toy causal model (See Figures 2b, 2c, 3b, and 3c; and Appendix Tables 4, 5, 7, and 8).

While we might not expect spurious signals to be as reliable out of domain, that does not mean that they will always fail. For example, while the associations between genre and sentiment learned from a dataset of book reviews might not hold in a dataset of kitchen appliances, but nevertheless hold in a dataset of audiobook reviews. In such settings, even though noising non-causal features would lead to models relying more on causal features, this may not result in better out-of-domain performance.

We also look at whether we really need to go through the process of collecting CAD (or human-annotated rationales) at all or if automated methods for generating "counterfactuals" might obtain similar gains in out-of-domain performance, as the former could be an expensive process. We experiment with state-of-the-art style transfer methods to convert *Positive* reviews into *Negative* and vice versa. Ideally, we would expect these methods to preserve a document's "content" while modifying the attributes that relate to sentiment (if they obtain perfect disentanglement in the feature space). Sentiment classifiers trained on original and *sentiment-flipped* reviews generated using style transfer methods often give better out-of-domain performance compared to training only on original data of same size (Table 2). However, models trained on CAD perform even better across all datasets, hinting at the value of human feedback.

## 5  CONCLUSION

While prior work offers promising clues to the benefits of CAD generated through human-in-the-loop mechanisms, previous work lacked formal frameworks for thinking about the technique, or comparisons to plausible alternatives. In this paper, through simple analysis on toy linear Gaussian models followed by a large-scale empirical investigation on sentiment analysis and NLI tasks, we formalize CAD and take some initial steps towards understanding its practical efficacy. Our analysis suggests that data corrupted by adding noise to rationale spans (analogous to adding noise to causal features) will degrade out-of-domain performance, while noise added to non-causal features may make models more robust out-of-domain. Our empirical study focuses on sentiment analysis and NLI and our findings remain consistent across datasets and models. Furthermore, the two tasks are subjectively very different as sentiment analysis requires a strong consideration of expressions of opinion than stated facts, whereas NLI is the opposite. We also show that models trained on the augmentation of original data and revised data generated by style transfer methods had better out-of-domain generalization in some cases compared to models trained on original data alone, but performed worse than models trained on CAD. In future work, we will look at how these findings generalize to other domains, including computer vision, and investigate the surprisingly low susceptibility of pre-trained transformers to spurious associations.

## ACKNOWLEDGEMENTS

The authors are grateful to NVIDIA for providing GPUs to conduct the experiments, Salesforce Research and Facebook AI for their financial support, and Sanket Mehta, Sina Fazelpour and Tejas Khot for our discussions and their valuable feedback.

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

# A  OLS ESTIMATION UNDER NOISY MEASUREMENT

## A.1  CAUSAL SETTING

Let the Gaussian SCM be defined as follows where the noise term for variable $x$ is defined as $u_x$:

$$
\begin{aligned}
z &= u_z, & u_z &\sim \mathcal{N}(0, \sigma_{u_z}^2) \\
x_1 &= bz + u_{x_1}, & u_{x_1} &\sim \mathcal{N}(0, \sigma_{u_{x1}}^2) \\
x_2 &= cz + u_{x_2}, & u_{x_2} &\sim \mathcal{N}(0, \sigma_{u_{x2}}^2) \\
y &= ax_1 + u_y, & u_y &\sim \mathcal{N}(0, \sigma_{u_y}^2).
\end{aligned}
\tag{4}
$$

$$
\begin{aligned}
\sigma_{x_1}^2 &= b^2 \sigma_{u_z}^2 + \sigma_{u_{x1}}^2 \\
\sigma_{x_2}^2 &= c^2 \sigma_{u_z}^2 + \sigma_{u_{x2}}^2 \\
\sigma_{x_1, x_2} &= bc\sigma_{u_z}^2 \\
\sigma_{x_1, y} &= ab^2 \sigma_{u_z}^2 + a\sigma_{u_{x1}}^2 \\
\sigma_{x_2, y} &= abc\sigma_{u_z}^2
\end{aligned}
\tag{5}
$$

Then if we were to solve the linear regression problem $y = x_1 \beta_1 + x_2 \beta_2 + \beta_0$, then using Eq. 1 we obtain the following values for $\beta_0^{ols}$, $\beta_1^{ols}$ and $\beta_2^{ols}$:

$$
\beta_1^{ols} = \frac{\sigma_{x_2}^2 \sigma_{x_1, y} - \sigma_{x_1, x_2} \sigma_{x_2, y}}{\sigma_{x_1}^2 \sigma_{x_2}^2 - \sigma_{x_1, x_2}^2} = \frac{(c^2 \sigma_{u_z}^2 + \sigma_{u_{x2}}^2)(ab^2 \sigma_{u_z}^2 + a\sigma_{u_{x1}}^2) - (bc\sigma_{u_z}^2)(abc\sigma_{u_z}^2)}{(b^2 \sigma_{u_z}^2 + \sigma_{u_{x1}}^2)(c^2 \sigma_{u_z}^2 + \sigma_{u_{x2}}^2) - b^2 c^2 \sigma_{u_z}^4}
\tag{6}
$$

$$
= a\frac{(b^2 \sigma_{u_z}^2 + \sigma_{u_{x1}}^2)(c^2 \sigma_{u_z}^2 + \sigma_{u_{x2}}^2) - b^2 c^2 \sigma_{u_z}^4}{(b^2 \sigma_{u_z}^2 + \sigma_{u_{x1}}^2)(c^2 \sigma_{u_z}^2 + \sigma_{u_{x2}}^2) - b^2 c^2 \sigma_{u_z}^4} = a
$$

$$
\beta_2^{ols} = \frac{\sigma_{x_1}^2 \sigma_{x_2, y} - \sigma_{x_1, x_2} \sigma_{x_1, y}}{\sigma_{x_1}^2 \sigma_{x_2}^2 - \sigma_{x_1, x_2}^2} = \frac{(b^2 \sigma_{u_z}^2 + \sigma_{u_{x1}}^2)(abc\sigma_{u_z}^2) - (bc\sigma_{u_z}^2)(ab^2 \sigma_{u_z}^2 + a\sigma_{u_{x1}}^2)}{(b^2 \sigma_{u_z}^2 + \sigma_{u_{x1}}^2)(c^2 \sigma_{u_z}^2 + \sigma_{u_{x2}}^2) - b^2 c^2 \sigma_{u_z}^4} = 0
\tag{7}
$$

However, if the setting is slightly different, and we observe a noisy version of $x_1$, given by $\widetilde{x_1}$:

$$
\widetilde{x_1} = x_1 + \epsilon_{x1}, \quad \epsilon_{x_1} \sim \mathcal{N}(0, \sigma_{\epsilon_{x1}}^2)
\tag{8}
$$

Since $\epsilon_{x_1} \perp\!\!\!\perp (x_1, x_2, y)$,

$$
\sigma_{\widetilde{x_1}}^2 = \mathrm{Var}[x_1 + \epsilon_{x1}] = b^2 \sigma_{u_z}^2 + \sigma_{u_{x1}}^2 + \sigma_{\epsilon_{x1}}^2
\tag{9}
$$

$$
\sigma_{\widetilde{x_1}, Y} = \sigma_{x_1, Y} = \mathrm{E}[(bz + u_{x_1})(ax_1 + u_y)] = ab^2 \sigma_{u_z}^2 + a\sigma_{u_{x1}}^2
\tag{10}
$$

$$
\sigma_{\widetilde{x_1}, x_2} = \sigma_{X_1, X_2} = bc\sigma_{u_z}^2
\tag{11}
$$

Plugging these values into Eq. 1 we get the OLS estimates $\widehat{\beta_1^{ols}}$ and $\widehat{\beta_2^{ols}}$ in the presence of observation noise on $X_1$:

$$
\widehat{\beta_1^{ols}} = \frac{\sigma_{x_2}^2 \sigma_{\widetilde{x_1}, y} - \sigma_{\widetilde{x_1}, x_2} \sigma_{x_2, y}}{\sigma_{\widetilde{x_1}}^2 \sigma_{x_2}^2 - \sigma_{\widetilde{x_1}, x_2}^2} = \frac{(c^2 \sigma_{u_z}^2 + \sigma_{u_{x2}}^2)(ab^2 \sigma_{u_z}^2 + a\sigma_{u_{x1}}^2) - (bc\sigma_{u_z}^2)(abc\sigma_{u_z}^2)}{(b^2 \sigma_{u_z}^2 + \sigma_{u_{x1}}^2 + \sigma_{\epsilon_{x1}}^2)(c^2 \sigma_{u_z}^2 + \sigma_{u_{x2}}^2) - b^2 c^2 \sigma_{u_z}^4}
$$

$$
= \frac{a(\sigma_{u_z}^2(b^2 \sigma_{u_{x2}}^2 + c^2 \sigma_{u_{x1}}^2) + \sigma_{u_{x1}}^2 \sigma_{u_{x2}}^2)}{\sigma_{u_z}^2(b^2 \sigma_{u_{x2}}^2 + c^2 \sigma_{u_{x1}}^2) + \sigma_{u_{x1}}^2 \sigma_{u_{x2}}^2 + \sigma_{\epsilon_{x1}}^2(c^2 \sigma_{u_z}^2 + \sigma_{u_{x2}}^2)}
$$

$$
= \frac{\beta_1^{ols}}{1 + \lambda_c}
$$

$$
\lambda_c = \frac{\sigma_{\epsilon_{x1}}^2(c^2 \sigma_{u_z}^2 + \sigma_{u_{x2}}^2)}{\sigma_{u_z}^2(b^2 \sigma_{u_{x2}}^2 + c^2 \sigma_{u_{x1}}^2) + \sigma_{u_{x1}}^2 \sigma_{u_{x2}}^2}
$$

$$
\widehat{\beta_2^{ols}} = \frac{\sigma_{\widetilde{x_1}}^2 \sigma_{x_2, y} - \sigma_{\widetilde{x_1}, x_2} \sigma_{\widetilde{x_1}, y}}{\sigma_{\widetilde{x_1}}^2 \sigma_{x_2}^2 - \sigma_{\widetilde{x_1}, x_2}^2} = \frac{(b^2 \sigma_{u_z}^2 + \sigma_{u_{x1}}^2 + \sigma_{\epsilon_{x1}}^2)abc\sigma_{u_z}^2 - (bc\sigma_{u_z}^2)(ab^2 \sigma_{u_z}^2 + a\sigma_{u_{x1}}^2)}{\sigma_{u_z}^2(b^2 \sigma_{u_{x2}}^2 + c^2 \sigma_{u_{x1}}^2) + \sigma_{u_{x1}}^2 \sigma_{u_{x2}}^2 + \sigma_{\epsilon_{x1}}^2(c^2 \sigma_{u_z}^2 + \sigma_{u_{x2}}^2)}
$$

$$
= \frac{acb\sigma_{\epsilon_{x1}}^2 \sigma_{u_z}^2}{\sigma_{u_z}^2(b^2 \sigma_{u_{x2}}^2 + c^2 \sigma_{u_{x1}}^2) + \sigma_{u_{x1}}^2 \sigma_{u_{x2}}^2 + \sigma_{\epsilon_{x1}}^2(c^2 \sigma_{u_z}^2 + \sigma_{u_{x2}}^2)}
\tag{12}
$$

As we can see $\lambda_c > 0$ and $\lambda_c \propto \sigma^2_{\epsilon_{x1}}$. This shows us that as $\sigma^2_{\epsilon_{x1}}$ increases, $|\widehat{\beta^{ols}_1}|$ (magnitude of the coefficient for $X_1$) decreases and $|\widehat{\beta^{ols}_2}|$ (magnitude of the coefficient for $X_2$) increases. $\lim_{\sigma^2_{\epsilon_{x1}} \to \infty} \widehat{\beta^{ols}_1} = 0$, and $\lim_{\sigma^2_{\epsilon_{x1}} \to \infty} \widehat{\beta^{ols}_2} = \frac{acb\sigma^2_{u_z}}{c^2\sigma^2_{u_z} + \sigma^2_{u_{x2}}}$.

## A.2 ANTICAUSAL SETTING

Once again we assume that each variable $V$ is a linear function of its parents $\text{Pa}(V)$. The noise terms are assumed to be Gaussian and are jointly independent.

$$
\begin{aligned}
z &= u_z, & u_z &\sim \mathcal{N}(0, \sigma^2_{u_z}) \\
q &= az + u_q, & u_q &\sim \mathcal{N}(0, \sigma^2_{u_q}) \\
y &= bz + u_y, & u_y &\sim \mathcal{N}(0, \sigma^2_{u_y}) \\
x_2 &= cq + u_{x_2}, & u_{x_1} &\sim \mathcal{N}(0, \sigma^2_{u_{x1}}) \\
x_1 &= dy + u_{x_1}, & u_{x_2} &\sim \mathcal{N}(0, \sigma^2_{u_{x2}})
\end{aligned}
\tag{13}
$$

$$
\begin{aligned}
\sigma^2_{x_1} &= d^2 b^2 \sigma^2_{u_z} + d^2 \sigma^2_{u_y} + \sigma^2_{u_{x1}} \\
\sigma^2_{x_2} &= c^2 a^2 \sigma^2_{u_z} + c^2 \sigma^2_{u_q} + \sigma^2_{u_{x2}} \\
\sigma_{x_1,x_2} &= abcd\sigma^2_{u_z} \\
\sigma_{x_1,y} &= db^2 \sigma^2_{u_z} + d\sigma^2_{u_y} \\
\sigma_{x_2,y} &= abc\sigma^2_{u_z}
\end{aligned}
\tag{14}
$$

If we were to solve the linear regression problem $y = x_1\beta_1 + x_2\beta_2 + \beta_0$, then using Eq. 1 we get the OLS estimates $\beta^{ols}_1$ and $\beta^{ols}_2$:

$$
\begin{aligned}
\beta^{ols}_1 &= \frac{\sigma^2_{x_2}\sigma_{x_1,y} - \sigma_{x_1,x_2}\sigma_{x_2,y}}{\sigma^2_{x_1}\sigma^2_{x_2} - \sigma_{x_1,x_2}{}^2} \\
&= \frac{(c^2 a^2 \sigma^2_{u_z} + c^2 \sigma^2_{u_q} + \sigma^2_{u_{x2}})(db^2 \sigma^2_{u_z} + d\sigma^2_{u_y}) - (abcd\sigma^2_{u_z})(abc\sigma^2_{u_z})}{(d^2 b^2 \sigma^2_{u_z} + d^2 \sigma^2_{u_y} + \sigma^2_{u_{x1}})(c^2 a^2 \sigma^2_{u_z} + c^2 \sigma^2_{u_q} + \sigma^2_{u_{x2}}) - (a^2 b^2 c^2 d^2 \sigma^2_{u_z}{}^2)} \\
&= \frac{d(a^2 c^2 \sigma^2_{u_z}\sigma^2_{u_y} + (c^2 \sigma^2_{u_q} + \sigma^2_{u_{x2}})(b^2 \sigma^2_{u_z} + \sigma^2_{u_y}))}{(d^2 b^2 \sigma^2_{u_z} + \sigma^2_{u_{x1}} + d^2 \sigma^2_{u_y})(\sigma^2_{u_{x2}} + c^2 \sigma^2_{u_q}) + (\sigma^2_{u_{x1}} + d^2 \sigma^2_{u_y})c^2 a^2 \sigma^2_{u_z}}
\end{aligned}
\tag{15}
$$

$$
\begin{aligned}
\beta^{ols}_2 &= \frac{\sigma^2_{x_1}\sigma_{x_2,y} - \sigma_{x_1,x_2}\sigma_{x_1,y}}{\sigma^2_{x_1}\sigma^2_{x_2} - \sigma_{x_1,x_2}{}^2} \\
&= \frac{(d^2 b^2 \sigma^2_{u_z} + d^2 \sigma^2_{u_y} + \sigma^2_{u_{x1}})(abc\sigma^2_{u_z}) - (abcd\sigma^2_{u_z})(db^2 \sigma^2_{u_z} + d\sigma^2_{u_y})}{(d^2 b^2 \sigma^2_{u_z} + d^2 \sigma^2_{u_y} + \sigma^2_{u_{x1}})(c^2 a^2 \sigma^2_{u_z} + c^2 \sigma^2_{u_q} + \sigma^2_{u_{x2}}) - (a^2 b^2 c^2 d^2 \sigma^2_{u_z}{}^2)} \\
&= \frac{abc\sigma^2_{u_z}\sigma^2_{u_{x1}}}{(d^2 b^2 \sigma^2_{u_z} + \sigma^2_{u_{x1}} + d^2 \sigma^2_{u_y})(\sigma^2_{u_{x2}} + c^2 \sigma^2_{u_q}) + (\sigma^2_{u_{x1}} + d^2 \sigma^2_{u_y})c^2 a^2 \sigma^2_{u_z}}
\end{aligned}
$$

However, if the setting is slightly different, and we observe a noisy version of $x_1$, given by $\widetilde{x_1}$:

$$
\widetilde{x_1} = x_1 + \epsilon_{x_1}, \qquad \epsilon_{x_1} \sim \mathcal{N}(0, \sigma^2_{\epsilon_{x1}})
\tag{16}
$$

Since $\epsilon_{x_1} \perp\!\!\!\perp x_2, y$, in order to obtain expressions for the OLS estimates $\widehat{\beta^{ols}_1}, \widehat{\beta^{ols}_2}$ in the presence of observation noise, in Eq. 15 we only need to replace $\sigma^2_{u_{x1}}$ with $\sigma^2_{u_{\widetilde{x1}}}$, which is given by:

$$
\sigma^2_{u_{\widetilde{x1}}} = \sigma^2_{u_{x1}} + \sigma^2_{\epsilon_{x1}}
\tag{17}
$$

$$\widehat{\beta_1^{ols}} = \frac{d(a^2c^2\sigma_{u_z}^2\sigma_{u_y}^2 + (c^2\sigma_{u_q}^2 + \sigma_{u_{x2}}^2)(b^2\sigma_{u_z}^2 + \sigma_{u_y}^2))}{(d^2b^2\sigma_{u_z}^2 + \sigma_{u_{\widetilde{x1}}}^2 + d^2\sigma_{u_y}^2)(\sigma_{u_{x2}}^2 + c^2\sigma_{u_q}^2) + (\sigma_{u_{\widetilde{x1}}}^2 + d^2\sigma_{u_y}^2)c^2a^2\sigma_{u_z}^2}$$

$$= \frac{d(a^2c^2\sigma_{u_z}^2\sigma_{u_y}^2 + (c^2\sigma_{u_q}^2 + \sigma_{u_{x2}}^2)(b^2\sigma_{u_z}^2 + \sigma_{u_y}^2))}{(d^2b^2\sigma_{u_z}^2 + (\sigma_{u_{x1}}^2 + \sigma_{\epsilon_{x1}}^2) + d^2\sigma_{u_y}^2)(\sigma_{u_{x2}}^2 + c^2\sigma_{u_q}^2) + ((\sigma_{u_{x1}}^2 + \sigma_{\epsilon_{x1}}^2) + d^2\sigma_{u_y}^2)c^2a^2\sigma_{u_z}^2} \tag{18}$$

$$\widehat{\beta_2^{ols}} = \frac{abc\sigma_{u_z}^2\sigma_{u_{\widetilde{x1}}}^2}{(d^2b^2\sigma_{u_z}^2 + \sigma_{u_{\widetilde{x1}}}^2 + d^2\sigma_{u_y}^2)(\sigma_{u_{x2}}^2 + c^2\sigma_{u_q}^2) + (\sigma_{u_{\widetilde{x1}}}^2 + d^2\sigma_{u_y}^2)c^2a^2\sigma_{u_z}^2}$$

$$= \frac{abc\sigma_{u_z}^2(\sigma_{u_{x1}}^2 + \sigma_{\epsilon_{x1}}^2)}{(d^2b^2\sigma_{u_z}^2 + (\sigma_{u_{x1}}^2 + \sigma_{\epsilon_{x1}}^2) + d^2\sigma_{u_y}^2)(\sigma_{u_{x2}}^2 + c^2\sigma_{u_q}^2) + ((\sigma_{u_{x1}}^2 + \sigma_{\epsilon_{x1}}^2) + d^2\sigma_{u_y}^2)c^2a^2\sigma_{u_z}^2} \tag{19}$$

$$\widehat{\beta_1^{ols}} = \frac{\beta_1^{ols}}{1 + \lambda_{ac}^{x_1}} \qquad\qquad \widehat{\beta_2^{ols}} = \frac{\beta_2^{ols}}{1 + \lambda_{ac}^{x_1}}\left[1 + \frac{\sigma_{\epsilon_{x1}}^2}{\sigma_{u_{x1}}^2}\right] \tag{20}$$

$$\lambda_{ac}^{x_1} = \frac{\sigma_{\epsilon_{x1}}^2(c^2a^2\sigma_{u_z}^2 + c^2\sigma_{u_q}^2 + \sigma_{u_{x2}}^2)}{(d^2b^2\sigma_{u_z}^2 + \sigma_{u_{x1}}^2 + d^2\sigma_{u_y}^2)(\sigma_{u_{x2}}^2 + c^2\sigma_{u_q}^2) + (\sigma_{u_{x1}}^2 + d^2\sigma_{u_y}^2)c^2a^2\sigma_{u_z}^2} \tag{21}$$

where $\lambda_{ac}^{x_1} > 0$ and $\lambda_{ac}^{x_1} \propto \sigma_{\epsilon_{x1}}^2$. Thus, as $\sigma_{\epsilon_{x1}}^2$ increases, $|\widehat{\beta_1^{ols}}|$ decreases. The asymptotic OLS estimates in the presence of infinite observational noise can be seen to be: $\lim_{\sigma_{\epsilon_{x1}}^2 \to \infty} \widehat{\beta_1^{ols}} = 0$ , where

as $\lim_{\sigma_{\epsilon_{x1}}^2 \to \infty} \widehat{\beta_2^{ols}} = \beta_2^{ols}\frac{((d^2b^2\sigma_{u_z}^2 + \sigma_{u_{x1}}^2 + d^2\sigma_{u_y}^2)(\sigma_{u_{x2}}^2 + c^2\sigma_{u_q}^2) + (\sigma_{u_{x1}}^2 + d^2\sigma_{u_y}^2)c^2a^2\sigma_{u_z}^2)}{(\sigma_{u_{x1}}^2(c^2a^2\sigma_{u_z}^2 + c^2\sigma_{u_q}^2 + \sigma_{u_{x2}}^2))}$.

Similarly, if we observe a noisy version of $X_2$, given by $\widetilde{X_2}$:
$$\widetilde{x_2} = x_2 + \epsilon_{x_2}, \qquad \epsilon_{x_2} \sim \mathcal{N}(0, \sigma_{\epsilon_{x2}}^2) \tag{22}$$

Since $\epsilon_{x_2} \perp\!\!\!\perp x_1, y$, in order to obtain expressions for the OLS estimates $\widehat{\beta_1^{ols}}, \widehat{\beta_2^{ols}}$ in the presence of observation noise on non-causal features, in Eq. 15 we only need to replace $\sigma_{u_{x2}}^2$ with $\sigma_{u_{\widetilde{x2}}}^2$, which is given by:
$$\sigma_{u_{\widetilde{x2}}}^2 = \sigma_{u_{x2}}^2 + \sigma_{\epsilon_{x2}}^2 \tag{23}$$

$$\widehat{\beta_1^{ols}} = \frac{d(a^2c^2\sigma_{u_z}^2\sigma_{u_y}^2 + (c^2\sigma_{u_q}^2 + \sigma_{u_{\widetilde{x2}}}^2)(b^2\sigma_{u_z}^2 + \sigma_{u_y}^2))}{(d^2b^2\sigma_{u_z}^2 + \sigma_{u_{x1}}^2 + d^2\sigma_{u_y}^2)(\sigma_{u_{\widetilde{x2}}}^2 + c^2\sigma_{u_q}^2) + (\sigma_{u_{x1}}^2 + d^2\sigma_{u_y}^2)c^2a^2\sigma_{u_z}^2}$$

$$= \frac{d(a^2c^2\sigma_{u_z}^2\sigma_{u_y}^2 + (c^2\sigma_{u_q}^2 + (\sigma_{u_{x2}}^2 + \sigma_{\epsilon_{x2}}^2))(b^2\sigma_{u_z}^2 + \sigma_{u_y}^2))}{(d^2b^2\sigma_{u_z}^2 + \sigma_{u_{x1}}^2 + d^2\sigma_{u_y}^2)((\sigma_{u_{x2}}^2 + \sigma_{\epsilon_{x2}}^2) + c^2\sigma_{u_q}^2) + (\sigma_{u_{x1}}^2 + d^2\sigma_{u_y}^2)c^2a^2\sigma_{u_z}^2} \tag{24}$$

$$\widehat{\beta_2^{ols}} = \frac{abc\sigma_{u_z}^2\sigma_{u_{x1}}^2}{(d^2b^2\sigma_{u_z}^2 + \sigma_{u_{x1}}^2 + d^2\sigma_{u_y}^2)(\sigma_{u_{\widetilde{x2}}}^2 + c^2\sigma_{u_q}^2) + (\sigma_{u_{x1}}^2 + d^2\sigma_{u_y}^2)c^2a^2\sigma_{u_z}^2} \tag{25}$$

$$= \frac{abc\sigma_{u_z}^2\sigma_{u_{x1}}^2}{(d^2b^2\sigma_{u_z}^2 + \sigma_{u_{x1}}^2 + d^2\sigma_{u_y}^2)((\sigma_{u_{x2}}^2 + \sigma_{\epsilon_{x2}}^2) + c^2\sigma_{u_q}^2) + (\sigma_{u_{x1}}^2 + d^2\sigma_{u_y}^2)c^2a^2\sigma_{u_z}^2} \tag{26}$$

$$\widehat{\beta_1^{ols}} = \frac{\beta_1^{ols}}{1 + \lambda_{ac}^{x_2}}\left[1 + \frac{\sigma_{\epsilon_{x2}}^2(b^2\sigma_{u_z}^2 + \sigma_{u_y}^2)}{a^2c^2\sigma_{u_z}^2\sigma_{u_y}^2 + (c^2\sigma_{u_q}^2 + \sigma_{u_{x2}}^2)(b^2\sigma_{u_z}^2 + \sigma_{u_y}^2)}\right] \qquad \widehat{\beta_2^{ols}} = \frac{\beta_2^{ols}}{1 + \lambda_{ac}^{x_2}}$$

$$\lambda_{ac}^{x_2} = \frac{\sigma_{\epsilon_{x2}}^2(d^2b^2\sigma_{u_z}^2 + \sigma_{u_{x1}}^2 + d^2\sigma_{u_y}^2)}{(d^2b^2\sigma_{u_z}^2 + \sigma_{u_{x1}}^2 + d^2\sigma_{u_y}^2)(\sigma_{u_{x2}}^2 + c^2\sigma_{u_q}^2) + (\sigma_{u_{x1}}^2 + d^2\sigma_{u_y}^2)c^2a^2\sigma_{u_z}^2} \tag{27}$$

where $\lambda_{ac}^{x_2} > 0$ and $\lambda_{ac}^{x_2} \propto \sigma_{\epsilon_{x2}}^2$. Thus, as $\sigma_{\epsilon_{x2}}^2$ increases, $|\widehat{\beta_1^{ols}}|$ increases. The asymptotic OLS estimates in the presence of infinite observational noise can be seen to be: $\lim\limits_{\sigma_{\epsilon_{x2}}^2 \to \infty} \widehat{\beta_2^{ols}} = 0$ , where

as $\lim\limits_{\sigma_{\epsilon_{x2}}^2 \to \infty} \widehat{\beta_1^{ols}} = \beta_1^{ols} \frac{(b^2\sigma_{u_z}^2 + \sigma_{u_y}^2)((d^2b^2\sigma_{u_z}^2 + \sigma_{u_{x1}}^2 + d^2\sigma_{u_y}^2)(\sigma_{u_{x2}}^2 + c^2\sigma_{u_q}^2) + (\sigma_{u_{x1}}^2 + d^2\sigma_{u_y}^2)c^2a^2\sigma_{u_z}^2)}{(a^2c^2\sigma_{u_z}^2\sigma_{u_y}^2 + (c^2\sigma_{u_q}^2 + \sigma_{u_{x2}}^2)(b^2\sigma_{u_z}^2 + \sigma_{u_y}^2))(d^2b^2\sigma_{u_z}^2 + \sigma_{u_{x1}}^2 + d^2\sigma_{u_y}^2)}$.

## B    MODEL IMPLEMENTATION DETAILS FOR SECTION 4

**Standard Methods**    We use `scikit-learn` (Pedregosa et al., 2011) implementations of SVMs and Naïve Bayes for sentiment analysis. We train these models on TF-IDF bag of words feature representations of the reviews (Jones, 1972). We identify parameters for both classifiers using grid search conducted over the validation set.

**BiLSTM**    We restrict the vocabulary to the most frequent $20k$ tokens, replacing out-of-vocabulary tokens by `UNK`. We fix the maximum input length at 330 tokens when training on reviews from Kaushik et al. (2020) and 2678 when doing so on Zaidan et al. (2007), and pad smaller reviews. Each token is represented by a randomly-initialized 300-dimensional embedding. Our model consists of a bidirectional LSTM (hidden dimension 128) with recurrent dropout (probability 0.5) and self attention following the embedding layer. We use the self attention implementation discussed in Lin et al. (2017) with hyperparameter values $d = 64$ and $r = 64$. To generate output, we feed this (fixed-length) representation through a fully-connected hidden layer (hidden dimension 32), and then a fully-connected output layer with softmax activation. We train all models for a maximum of 20 epochs using Adam (Kingma & Ba, 2015), with a learning rate of $1e-4$ and a batch size of 16. We apply early stopping when validation loss does not decrease for 5 epochs.

**Pretrained Transformers**    We use off-the-shelf uncased BERT Base and Longformer Base models (Wolf et al., 2019), fine-tuning for each task. We used BERT for experiments on the smaller IMDb dataset used by Kaushik et al. (2020) (with a maximum review length of 330 tokens) and Longformer for the dataset presented by Zaidan et al. (2007) (with maximum review length of 2678). To account for BERT's sub-word tokenization, we set the maximum token length is set at 350 for sentiment analysis and 50 for NLI. In case of Longformer, that is 3072.[3] We fine-tune BERT up to 20 epochs with same early stopping criteria as for BiLSTM, using the BERT Adam optimizer with a batch size of 16 (to fit on a 16GB Tesla V-100 GPU). We found learning rates of $5e-5$ and $1e-5$ to work best for sentiment analysis and NLI respectively. We fine-tune Longformer for 10 epochs with early stopping, using a batch size of 8 (to fit on 64GB of GPU memory).

**Style Transfer Methods**    For Hu et al. (2017),[4] Sudhakar et al. (2019),[5] and Madaan et al. (2020),[6] we found the default hyperparameters used by the authors to work best on our task. In case of Li et al. (2018),[7] we followed the training schedule presented in the paper. However, since the paper does not present results on IMDb reviews, we experimented with multiple values of the *salience ratio*, and used a salience ratio of 5.5 for our downstream task based on transfer accuracy and BLEU scores achieved on the validation set. For all style transfer methods, we experimented with multiple sequence lengths, and found that models worked best on sentence level (versus review-level) data, with sequence length of 30, truncating longer sentences in the process. For each review, we passed individual sentences through each model and reconstructed whole reviews by joining the resulting *sentiment-flipped* sentences.

---

[3]Longformer is better suited to work on longer texts compared to BERT. Maximum length of a review in Zaidan et al. is 2678 tokens whereas in Kaushik et al. is only 330 tokens.

[4]https://github.com/asyml/texar/tree/master/examples/text_style_transfer

[5]https://github.com/agaralabs/transformer-drg-style-transfer

[6]https://github.com/tag-and-generate/

[7]https://github.com/lijuncen/Sentiment-and-Style-Transfer

# C  FULL RESULTS CORRESPONDING TO NOISE INJECTION

Table 3: Accuracy of various sentiment analysis classifiers trained on $1.7k$ original reviews from Kaushik et al. (2020) as noise is injected on *rationales/non-rationales* identified via human feedback.

| Dataset | Percent noise in rationales | | | | | | | | | | |
|---|---|---|---|---|---|---|---|---|---|---|---|
| | **SVM** | | | | | | | | | | |
| | 0 | 10 | 20 | 30 | 40 | 50 | 60 | 70 | 80 | 90 | 100 |
| In-sample test | 87.8 | 88.2 | 85.7 | 86.9 | 86.9 | 84.5 | 83.3 | 81.6 | 80 | 79.2 | 76.7 |
| CRD | 51.8 | 47.3 | 45.7 | 42.9 | 39.2 | 33.5 | 28.2 | 25.7 | 24.1 | 19.6 | 17.1 |
| Amazon | 73.2 | 72.2 | 71.3 | 69.4 | 67.3 | 63.7 | 63.7 | 58.2 | 57 | 50.1 | 46.5 |
| Semeval | 62.5 | 62.2 | 61.9 | 61.1 | 60.9 | 58.3 | 57.1 | 55.4 | 54.5 | 51.3 | 50.1 |
| Yelp | 79.9 | 79 | 77.7 | 76.7 | 74.1 | 71.4 | 69 | 65.5 | 62.4 | 55.8 | 51.5 |
| | **BiLSTM with Self Attention** | | | | | | | | | | |
| In-sample test | 81.5 | 78.8 | 77.6 | 76.7 | 75.3 | 75.2 | 74.5 | 72.8 | 67.3 | 64.2 | 63.8 |
| CRD | 49.4 | 49.3 | 46.3 | 45.1 | 39.5 | 38.1 | 38.9 | 38.7 | 32.6 | 32.6 | 29.7 |
| Amazon | 65.4 | 69.1 | 68.5 | 66.6 | 63.2 | 63.9 | 58.8 | 50.6 | 50.6 | 47.1 | 44.2 |
| Semeval | 59.3 | 59.8 | 57.6 | 56.4 | 58.6 | 56.6 | 55.3 | 54.3 | 54.3 | 52.3 | 50 |
| Yelp | 71.2 | 70.8 | 67.4 | 65.9 | 65.3 | 64.1 | 63.4 | 60.1 | 62.4 | 49.8 | 46.4 |
| | **BERT** | | | | | | | | | | |
| In-sample test | 87.4 | 87.4 | 86.5 | 85.7 | 85.3 | 84.3 | 83.6 | 81 | 76.6 | 71 | 69 |
| CRD | 82.2 | 78.1 | 78.4 | 75.4 | 67.6 | 67.5 | 65.5 | 53.9 | 42.7 | 36.2 | 31.8 |
| Amazon | 76.2 | 75.5 | 75.1 | 74.2 | 73.5 | 73 | 72.5 | 70.7 | 63.4 | 57.8 | 56.1 |
| Semeval | 76.4 | 69.7 | 66.9 | 69.8 | 67.8 | 67.4 | 66.8 | 65.5 | 62.2 | 54.9 | 52.6 |
| Yelp | 83.7 | 82.5 | 82 | 81.5 | 80.9 | 80.2 | 79.9 | 75.6 | 64.3 | 54.6 | 52.3 |
| Dataset | Percent noise in non-rationales | | | | | | | | | | |
| | **SVM** | | | | | | | | | | |
| In-sample test | 87.8 | 88.6 | 89 | 86.9 | 85.3 | 82.4 | 86.5 | 83.7 | 82 | 81.6 | 78 |
| CRD | 51.8 | 55.9 | 53.5 | 57.1 | 58.8 | 63.7 | 63.3 | 65.7 | 70.2 | 73.9 | 74.3 |
| Amazon | 73.2 | 74.9 | 75.3 | 77.3 | 75.8 | 76.6 | 76.5 | 77.4 | 75.5 | 75.4 | 76.9 |
| Semeval | 62.5 | 63.3 | 62.7 | 64.3 | 64.3 | 65.6 | 66 | 65.8 | 65 | 66.4 | 66.4 |
| Yelp | 79.9 | 80.9 | 80.1 | 82.2 | 83.6 | 84.1 | 83.5 | 83.4 | 82.7 | 82.1 | 81.4 |
| | **BiLSTM with Self Attention** | | | | | | | | | | |
| In-sample test | 81.5 | 77.5 | 77 | 75.9 | 75.4 | 75.2 | 75.1 | 73.8 | 73 | 72.4 | 71.7 |
| CRD | 49.4 | 53.1 | 56.25 | 56.6 | 57.5 | 58.4 | 58.6 | 60.3 | 61.5 | 65.5 | 66.1 |
| Amazon | 65.4 | 66.5 | 66.6 | 66.6 | 67.6 | 67.7 | 68.3 | 68.6 | 68.8 | 68.5 | 68.4 |
| Semeval | 59.3 | 58.6 | 58.9 | 59.3 | 58.1 | 57.5 | 59.2 | 59.5 | 59.8 | 59.5 | 58 |
| Yelp | 71.2 | 74.7 | 72.5 | 73.3 | 73.9 | 73.6 | 72.2 | 74.3 | 73.7 | 75.6 | 75.4 |
| | **BERT** | | | | | | | | | | |
| In-sample test | 87.4 | 88.2 | 87 | 86.9 | 87 | 85.8 | 83.6 | 78.9 | 72.5 | 72.1 | 71.3 |
| CRD | 82.2 | 92.8 | 92.8 | 92.3 | 93.1 | 92.8 | 89.8 | 88.6 | 84.5 | 81.3 | 81 |
| Amazon | 76.2 | 78.6 | 78.9 | 79.2 | 75.1 | 71.7 | 67.6 | 65.3 | 65.2 | 63.7 | 61.8 |
| Semeval | 76.4 | 74.6 | 76.3 | 75.8 | 70.9 | 62.1 | 64.8 | 63.3 | 60.8 | 58.7 | 58.7 |
| Yelp | 83.7 | 85.4 | 85.3 | 85.1 | 82.1 | 78.3 | 77.2 | 76.2 | 74.3 | 71.6 | 70.1 |

Table 4: Accuracy of various sentiment analysis classifiers trained on $1.7k$ original reviews from Kaushik et al. (2020) as noise is injected on *rationales/non-rationales* identified via Attention masks.

| Dataset | Percent noise in rationales | | | | | | | | | | |
|---|---|---|---|---|---|---|---|---|---|---|---|
| | **SVM** | | | | | | | | | | |
| | 0 | 10 | 20 | 30 | 40 | 50 | 60 | 70 | 80 | 90 | 100 |
| In-sample test | 87.8 | 85 | 85.9 | 86.3 | 86.3 | 85.2 | 84.6 | 86.3 | 83.6 | 84.2 | 83.6 |
| CRD | 51.8 | 50.6 | 51.8 | 52 | 51.8 | 50 | 50.6 | 48.6 | 48.6 | 47.5 | 46.1 |
| Amazon | 73.2 | 74.3 | 73.4 | 72.8 | 72.8 | 72.9 | 72 | 72.3 | 71.1 | 72 | 70.3 |
| Semeval | 62.5 | 62.8 | 62.9 | 61.8 | 62.5 | 61.9 | 61.4 | 60.7 | 61.1 | 60.6 | 60.1 |
| Yelp | 79.9 | 80.1 | 79.3 | 78.7 | 78.9 | 78.5 | 77.8 | 77.5 | 77.8 | 76.2 | 75.9 |
| | **BiLSTM with Self Attention** | | | | | | | | | | |
| In-sample test | 81.5 | 78.8 | 78.6 | 78.3 | 78.2 | 76.2 | 77.3 | 76.8 | 71.8 | 73.2 | 74.2 |
| CRD | 49.4 | 53.3 | 50 | 53.4 | 52.4 | 49.7 | 49.2 | 47.4 | 47.7 | 47 | 44.1 |
| Amazon | 65.4 | 66.8 | 71 | 64.7 | 60.7 | 61.7 | 65.2 | 64.6 | 51.6 | 57.1 | 66.4 |
| Semeval | 59.3 | 59.5 | 60.1 | 57.4 | 55.9 | 57.2 | 52.2 | 57.6 | 51.5 | 51.8 | 56.1 |
| Yelp | 71.2 | 72.3 | 74.2 | 69.6 | 70.5 | 67.3 | 70.7 | 72.8 | 62.8 | 65 | 66.2 |
| | **BERT** | | | | | | | | | | |
| In-sample test | 87.4 | 93 | 90.8 | 90.3 | 90.6 | 91.2 | 90.3 | 90.4 | 90.7 | 90.6 | 90.3 |
| CRD | 82.2 | 91.2 | 92 | 90.8 | 90.8 | 90.9 | 90.3 | 90.9 | 90.2 | 89.8 | 90.4 |
| Amazon | 76.2 | 77.3 | 79.1 | 78.7 | 79.8 | 79.1 | 79.8 | 79.5 | 79.2 | 78.9 | 79.3 |
| Semeval | 76.4 | 71.4 | 73.5 | 73.2 | 74.4 | 76.1 | 77.6 | 79.8 | 78.4 | 79.2 | 77.8 |
| Yelp | 83.7 | 83.5 | 85.4 | 84.9 | 86 | 85.7 | 85.9 | 85.6 | 85.5 | 85.4 | 68.9 |
| Dataset | Percent noise in non-rationales | | | | | | | | | | |
| | **SVM** | | | | | | | | | | |
| In-sample test | 87.8 | 85 | 85.7 | 84.8 | 85 | 84 | 83.6 | 84.6 | 80.7 | 81.1 | 77.3 |
| CRD | 51.8 | 50.4 | 52.2 | 53.9 | 50.2 | 50.8 | 52.9 | 54.1 | 54.1 | 56.8 | 56.4 |
| Amazon | 73.2 | 73.5 | 75.3 | 74.3 | 76.2 | 73.9 | 73.4 | 73.6 | 71 | 70 | 67.8 |
| Semeval | 62.5 | 62.6 | 63.7 | 63.7 | 63.1 | 62.6 | 63.5 | 61.5 | 62.1 | 62 | 59.9 |
| Yelp | 79.9 | 79.8 | 80.9 | 81.7 | 80.9 | 80.5 | 80 | 80.1 | 78.5 | 77.5 | 74.4 |
| | **BiLSTM with Self Attention** | | | | | | | | | | |
| In-sample test | 81.5 | 77.6 | 76 | 77.1 | 77.3 | 75.4 | 73.7 | 67.9 | 68.6 | 54.2 | 52.3 |
| CRD | 49.4 | 53.1 | 52.1 | 52.1 | 65 | 54.1 | 51.9 | 53.4 | 55 | 52.3 | 51.6 |
| Amazon | 65.4 | 63.7 | 65.7 | 64 | 58.8 | 65.5 | 60.3 | 58.7 | 61 | 58.1 | 56.2 |
| Semeval | 59.3 | 54.8 | 58.4 | 57.3 | 60.7 | 56.8 | 55.2 | 54 | 51.2 | 50 | 49.9 |
| Yelp | 71.2 | 72 | 73.6 | 70.2 | 61.3 | 71.5 | 68.4 | 64.9 | 66.3 | 58.2 | 55.8 |
| | **BERT** | | | | | | | | | | |
| In-sample test | 87.4 | 86.9 | 86.7 | 85.3 | 84 | 81.9 | 80.6 | 74 | 74 | 73 | 67.2 |
| CRD | 82.2 | 92.3 | 92.4 | 92.1 | 90 | 86.8 | 83 | 73.2 | 77.7 | 72.5 | 68.5 |
| Amazon | 76.2 | 79.5 | 78.5 | 77.9 | 69.2 | 67.4 | 58.1 | 55.9 | 53.5 | 55.8 | 52.6 |
| Semeval | 76.4 | 76.5 | 75.7 | 77.1 | 65.7 | 61.8 | 54.6 | 58.8 | 51.8 | 54 | 50.8 |
| Yelp | 83.7 | 85.8 | 85 | 85.5 | 79.3 | 78.7 | 67.8 | 66.5 | 59.5 | 63.2 | 57.5 |

Table 5: Accuracy of various sentiment analysis classifiers trained on $1.7k$ original reviews from Kaushik et al. (2020) as noise is injected on *rationales/non-rationales* identified via Allen NLP Saliency Interpreter.

| Dataset | Percent noise in rationales | | | | | | | | | | |
|---|---|---|---|---|---|---|---|---|---|---|---|
| | SVM | | | | | | | | | | |
| | 0 | 10 | 20 | 30 | 40 | 50 | 60 | 70 | 80 | 90 | 100 |
| In-sample test | 87.8 | 85.1 | 85.4 | 85.1 | 85.1 | 83.9 | 82.5 | 82.8 | 81.8 | 80 | 77.5 |
| CRD | 51.8 | 51.2 | 52.5 | 51.1 | 51 | 50.1 | 49.5 | 46.6 | 43.7 | 42.1 | 40.5 |
| Amazon | 73.2 | 73.4 | 73.65 | 73.2 | 72.7 | 72.9 | 71.8 | 72.1 | 70.5 | 69.6 | 68.9 |
| Semeval | 62.5 | 62.8 | 62.5 | 62.4 | 61.9 | 61.2 | 60.7 | 60.5 | 59.6 | 58.4 | 57.9 |
| Yelp | 79.9 | 79.8 | 79.7 | 79.1 | 78.7 | 78.2 | 78.1 | 76.6 | 75.1 | 74.1 | 72.2 |
| | BiLSTM with Self Attention | | | | | | | | | | |
| In-sample test | 81.5 | 82.3 | 83.5 | 80.4 | 78.2 | 81.9 | 80.6 | 77.8 | 79.2 | 76 | 77 |
| CRD | 49.4 | 48.2 | 48.6 | 51.2 | 48.6 | 47.3 | 47.1 | 46.9 | 44.3 | 42.6 | 37.9 |
| Amazon | 65.4 | 46.6 | 72.8 | 66.9 | 49.7 | 55.4 | 53.7 | 68.5 | 54.7 | 49.8 | 51.8 |
| Semeval | 59.3 | 42.1 | 49.5 | 56.2 | 54.7 | 52.7 | 53.7 | 50.1 | 51.2 | 50.2 | 50 |
| Yelp | 71.2 | 69 | 73.3 | 73.2 | 67.8 | 69.2 | 69.5 | 68.8 | 67 | 54.4 | 56.9 |
| | BERT | | | | | | | | | | |
| In-sample test | 87.4 | 91.1 | 90.6 | 90 | 88 | 89.1 | 87.4 | 86.3 | 83.6 | 84.5 | 81.6 |
| CRD | 82.2 | 93.4 | 92.3 | 91.9 | 90.3 | 90.2 | 87.7 | 83.8 | 78 | 79.3 | 70 |
| Amazon | 76.2 | 82.4 | 81.3 | 79.8 | 77.2 | 77.6 | 77.8 | 75.6 | 69.7 | 69.4 | 73.6 |
| Semeval | 76.4 | 82.6 | 82.8 | 81.3 | 79.2 | 78.1 | 76.7 | 74.7 | 67.4 | 65.8 | 67.4 |
| Yelp | 83.7 | 88.3 | 88.8 | 88.5 | 87.8 | 88.1 | 87 | 86.2 | 84.4 | 83.3 | 82.7 |
| Dataset | Percent noise in non-rationales | | | | | | | | | | |
| | SVM | | | | | | | | | | |
| In-sample test | 87.8 | 85.9 | 85.7 | 86.9 | 83.6 | 86.9 | 85.9 | 83.2 | 85 | 81.8 | 79.1 |
| CRD | 51.8 | 52.3 | 53.7 | 53.9 | 56.8 | 55.3 | 53.5 | 54.3 | 58 | 60 | 61.5 |
| Amazon | 73.2 | 73.9 | 74.1 | 71.8 | 73.6 | 72.5 | 73.8 | 72.6 | 70.6 | 70.6 | 70.8 |
| Semeval | 62.5 | 62.7 | 62.8 | 61.3 | 62.7 | 62 | 61.9 | 63.2 | 62.3 | 62.4 | 63.6 |
| Yelp | 79.9 | 79.8 | 79.8 | 81.4 | 81 | 80.7 | 81 | 80.5 | 80.3 | 79.8 | 78.6 |
| | BiLSTM with Self Attention | | | | | | | | | | |
| In-sample test | 81.5 | 81 | 81.7 | 80.8 | 79.8 | 78 | 75.6 | 73 | 70.4 | 51 | 50 |
| CRD | 49.4 | 49 | 49.8 | 48 | 47.9 | 51.6 | 46.7 | 53.3 | 50.2 | 51.6 | 48.4 |
| Amazon | 65.4 | 65.3 | 64.9 | 62.7 | 63.3 | 65.3 | 67.1 | 65.3 | 64 | 58.3 | 41.8 |
| Semeval | 59.3 | 55 | 61.3 | 50.1 | 54.6 | 58.5 | 55.2 | 55.7 | 49.4 | 49.6 | 44 |
| Yelp | 71.2 | 73.8 | 75.1 | 71.4 | 74.1 | 73.4 | 74.5 | 72.5 | 66.9 | 55.9 | 53.6 |
| | BERT | | | | | | | | | | |
| In-sample test | 87.4 | 90.5 | 89.1 | 88.6 | 80.6 | 75.1 | 70.1 | 63.7 | 53.8 | 54.1 | 53.1 |
| CRD | 82.2 | 92.1 | 92.2 | 91.3 | 79.9 | 73.3 | 67.1 | 59.2 | 50.1 | 49.8 | 49.6 |
| Amazon | 76.2 | 77.5 | 79.2 | 77.3 | 69 | 65.9 | 61.1 | 61.7 | 52.9 | 52.7 | 51.4 |
| Semeval | 76.4 | 82 | 83.6 | 83.1 | 78.1 | 77.9 | 71.1 | 69.7 | 55.9 | 56.5 | 51.8 |
| Yelp | 83.7 | 88 | 87.4 | 87.8 | 76.8 | 73 | 66.9 | 66.3 | 55.3 | 54.4 | 53.1 |

Table 6: Accuracy of various sentiment analysis classifiers trained on reviews from Zaidan et al. (2007) as noise is injected on *rationales/non-rationales* identified via human feedback.

| Dataset | Percent noise in rationales | | | | | | | | | | |
|---|---|---|---|---|---|---|---|---|---|---|---|
| | SVM | | | | | | | | | | |
| | 0 | 10 | 20 | 30 | 40 | 50 | 60 | 70 | 80 | 90 | 100 |
| In-sample test | 87.5 | 86.2 | 85.5 | 85 | 84.5 | 83.3 | 82.5 | 81.1 | 78.9 | 77.5 | 76.5 |
| CRD | 46.1 | 45.6 | 44.4 | 43.7 | 44.1 | 41.2 | 38.8 | 36 | 34.4 | 33.1 | 30.9 |
| Amazon | 68.6 | 67.1 | 65.1 | 64.2 | 62.2 | 60.4 | 57.9 | 50.5 | 54.9 | 53.5 | 51.8 |
| Semeval | 56.7 | 56.1 | 55.4 | 54.8 | 54.1 | 53.5 | 52.7 | 52 | 51.6 | 50.8 | 50.4 |
| Yelp | 76.2 | 75 | 73.5 | 72 | 70.2 | 68.8 | 66.6 | 65.1 | 63.3 | 61.1 | 59.3 |
| | BiLSTM with Self Attention | | | | | | | | | | |
| In-sample test | 80.3 | 82.1 | 83.2 | 81.3 | 78.4 | 71.1 | 78.8 | 77.4 | 76.9 | 77.4 | 75.5 |
| CRD | 49.2 | 50.6 | 51 | 48.8 | 48 | 49.6 | 49.4 | 48.8 | 48.8 | 47.5 | 48.4 |
| Amazon | 50 | 50.5 | 49.4 | 49.7 | 49.8 | 49.7 | 49.7 | 49.7 | 49.6 | 49.5 | 49.4 |
| Semeval | 50 | 50 | 50 | 50 | 50 | 50 | 50 | 50 | 50 | 50 | 50 |
| Yelp | 50.5 | 50 | 53.1 | 52.1 | 50.5 | 50.2 | 50.1 | 50 | 50 | 50.2 | 50.1 |
| | Longformer | | | | | | | | | | |
| In-sample test | 97.5 | 96.7 | 94 | 90.5 | 88.3 | 78.9 | 81.4 | 72.6 | 79.4 | 78.7 | 83.5 |
| CRD | 93.4 | 93.6 | 87.5 | 85.4 | 84.2 | 64.1 | 61.5 | 54.2 | 52.7 | 50.3 | 48 |
| Amazon | 81.8 | 77.9 | 65.3 | 65.7 | 64.7 | 63.6 | 61.9 | 62.1 | 61.3 | 60.6 | 57.9 |
| Semeval | 80.3 | 74.9 | 64 | 66.9 | 71.6 | 61.3 | 58.4 | 56.7 | 58.9 | 62.1 | 58.6 |
| Yelp | 88.6 | 85.8 | 77.7 | 74.6 | 72.5 | 68.4 | 66.5 | 64.8 | 64.3 | 64.9 | 62.2 |
| Dataset | Percent noise in non-rationales | | | | | | | | | | |
| | SVM | | | | | | | | | | |
| In-sample test | 87.5 | 85.5 | 86 | 83 | 82 | 83 | 81 | 80.5 | 75.5 | 60 | 50 |
| CRD | 46.1 | 46.1 | 49 | 49.4 | 57.1 | 55.5 | 58.4 | 58.4 | 56.5 | 56.3 | 54 |
| Amazon | 68.6 | 67.7 | 68 | 67.2 | 69.4 | 69 | 69.7 | 68.9 | 69.2 | 64.9 | 62.3 |
| Semeval | 56.7 | 56.9 | 57.5 | 57.4 | 58.3 | 57.6 | 58.8 | 59.4 | 59.3 | 57.4 | 56.3 |
| Yelp | 76.2 | 76.1 | 76.9 | 75.9 | 77 | 77.4 | 75.2 | 74.1 | 73.3 | 68.5 | 61.6 |
| | BiLSTM with Self Attention | | | | | | | | | | |
| In-sample test | 80.3 | 80.8 | 79.8 | 75.2 | 75 | 62.5 | 62 | 57.7 | 56.7 | 58.7 | 57.7 |
| CRD | 49.2 | 50 | 51.1 | 50.8 | 52.9 | 53.9 | 58.6 | 58.6 | 60 | 60.4 | 60.8 |
| Amazon | 50 | 50 | 50.7 | 50.7 | 50.9 | 52.2 | 52.3 | 53.2 | 55 | 55.1 | 56.7 |
| Semeval | 50 | 50 | 50 | 50 | 50 | 51 | 51.8 | 52.7 | 53.5 | 53.8 | 53.9 |
| Yelp | 50.5 | 50.4 | 52.7 | 52.9 | 52.9 | 55.2 | 58 | 58.9 | 64.6 | 64.6 | 70 |
| | Longformer | | | | | | | | | | |
| In-sample test | 97.5 | 97.9 | 98.1 | 97.4 | 94.8 | 93.4 | 86.4 | 82.3 | 76.3 | 77.4 | 80.2 |
| CRD | 93.4 | 94.7 | 94.1 | 91.8 | 91.4 | 91.8 | 88 | 83.4 | 83.7 | 83.6 | 83.4 |
| Amazon | 81.8 | 79 | 80 | 81.5 | 83.2 | 84.2 | 84.1 | 76.3 | 78.5 | 79.4 | 76.9 |
| Semeval | 80.3 | 79.4 | 77.2 | 80.6 | 80.6 | 84.6 | 85.3 | 71.8 | 79.9 | 83.7 | 76.6 |
| Yelp | 88.6 | 85.3 | 86.4 | 89 | 89.5 | 89.9 | 89.9 | 86.2 | 86.5 | 86.4 | 84.7 |

Table 7: Accuracy of various sentiment analysis classifiers trained on reviews from Zaidan et al. (2007) as noise is injected on *rationales/non-rationales* identified via Attention masks.

| Dataset | Percent noise in rationales | | | | | | | | | | |
|---|---|---|---|---|---|---|---|---|---|---|---|
| **SVM** | | | | | | | | | | | |
| | 0 | 10 | 20 | 30 | 40 | 50 | 60 | 70 | 80 | 90 | 100 |
| In-sample test | 87.5 | 85 | 84.5 | 84 | 82.5 | 83 | 81 | 80 | 77.5 | 75.5 | 75.5 |
| CRD | 46.1 | 51 | 50.6 | 52 | 51.8 | 52.3 | 52.3 | 51.8 | 50.2 | 49.8 | 49.8 |
| Amazon | 68.6 | 68.1 | 67.1 | 66.8 | 66.9 | 66.5 | 66.2 | 65.4 | 66.1 | 66.6 | 65.7 |
| Semeval | 56.7 | 56.6 | 56.3 | 56.4 | 56.2 | 56.4 | 56.4 | 56.2 | 56.8 | 56.4 | 56.4 |
| Yelp | 76.2 | 76.1 | 76 | 76.2 | 76.4 | 76.5 | 76.9 | 76.9 | 76.7 | 76.9 | 76.5 |
| **BiLSTM with Self Attention** | | | | | | | | | | | |
| In-sample test | 80.3 | 78.8 | 77.9 | 77.9 | 78.8 | 67.3 | 65.9 | 63.9 | 62 | 65.4 | 58.7 |
| CRD | 49.2 | 49.4 | 50.2 | 50.2 | 52.1 | 51 | 52.1 | 52.3 | 56.3 | 51.8 | 54.7 |
| Amazon | 50 | 49.7 | 49.9 | 49.9 | 50.4 | 50.2 | 51 | 51.7 | 51.1 | 50.7 | 50.7 |
| Semeval | 50 | 50 | 50 | 50 | 50 | 50 | 50 | 50.2 | 50.1 | 50 | 50.1 |
| Yelp | 50.5 | 50.1 | 50.5 | 50.5 | 52.1 | 52.4 | 56.1 | 54.9 | 54.9 | 52.2 | 54.9 |
| **Longformer** | | | | | | | | | | | |
| In-sample test | 97.5 | 97.3 | 97 | 96.5 | 88.3 | 94 | 93.8 | 91.2 | 91.5 | 87.2 | 84 |
| CRD | 93.4 | 93.5 | 93.1 | 92.8 | 91.7 | 91.8 | 90.7 | 88 | 87.5 | 83.7 | 80.8 |
| Amazon | 81.8 | 76.3 | 69.5 | 75.4 | 70.4 | 64.5 | 66.3 | 60.8 | 64.7 | 57.3 | 55.3 |
| Semeval | 80.3 | 73 | 67.2 | 75.1 | 69.6 | 61.5 | 67 | 58.8 | 67.6 | 56.4 | 55.3 |
| Yelp | 88.6 | 85.1 | 79.3 | 83.9 | 79.8 | 75.4 | 76.8 | 69.1 | 75.4 | 65.7 | 61 |
| Dataset | Percent noise in non-rationales | | | | | | | | | | |
| **SVM** | | | | | | | | | | | |
| In-sample test | 87.5 | 87 | 86.5 | 87.5 | 81 | 82.5 | 73 | 52 | 50 | 50 | 50 |
| CRD | 46.1 | 50.4 | 49.6 | 48.6 | 50 | 46.9 | 50.6 | 49.6 | 50.4 | 50.2 | 50.2 |
| Amazon | 68.6 | 66.7 | 66.8 | 64.1 | 65.9 | 63.2 | 62.2 | 60 | 57.8 | 56.2 | 56.3 |
| Semeval | 56.7 | 56.3 | 56.8 | 55.9 | 56.7 | 55 | 54.2 | 53.8 | 51.8 | 51.1 | 51 |
| Yelp | 76.2 | 74.8 | 74.2 | 71.1 | 71 | 64.9 | 59.7 | 55.2 | 52.3 | 51 | 50 |
| **BiLSTM with Self Attention** | | | | | | | | | | | |
| In-sample test | 80.3 | 79.8 | 81.3 | 78.4 | 63.5 | 67.3 | 49.5 | 49 | 48.1 | 48.4 | 48.1 |
| CRD | 49.2 | 51.4 | 51.4 | 54.5 | 49.8 | 49.4 | 49.6 | 49.4 | 49.4 | 49.4 | 49.4 |
| Amazon | 50 | 49.9 | 50.6 | 50.4 | 50.1 | 49.7 | 49.6 | 49.5 | 49.5 | 49.5 | 49.5 |
| Semeval | 50 | 50 | 50 | 50.2 | 50 | 50 | 50 | 50 | 50 | 50 | 50 |
| Yelp | 50.5 | 52.3 | 52.7 | 56.9 | 51 | 50.4 | 50 | 50 | 50 | 50 | 50 |
| **Longformer** | | | | | | | | | | | |
| In-sample test | 97.5 | 98.2 | 97.8 | 95 | 90.2 | 83.3 | 67.3 | 62.8 | 69.3 | 64.2 | 52.8 |
| CRD | 93.4 | 93.6 | 93.5 | 88.8 | 83.1 | 76.5 | 67.8 | 69.7 | 77.6 | 54.5 | 51.4 |
| Amazon | 81.8 | 81.6 | 97.8 | 95 | 90.2 | 83.3 | 67.3 | 62.8 | 79.3 | 64.2 | 52.8 |
| Semeval | 80.3 | 74.8 | 70.3 | 79.1 | 79 | 78.9 | 69.5 | 67.9 | 64 | 63.3 | 58.6 |
| Yelp | 88.6 | 83.9 | 83.1 | 89.5 | 90.2 | 89.7 | 87.6 | 83 | 78.8 | 62.4 | 59.4 |

Table 8: Accuracy of various sentiment analysis classifiers trained on reviews from Zaidan et al. (2007) as noise is injected on *rationales/non-rationales* identified via Allen NLP Saliency interpreter.

| Dataset | Percent rationales tokens replaced by noise | | | | | | | | | | |
|---|---|---|---|---|---|---|---|---|---|---|---|
| | SVM | | | | | | | | | | |
| | 0 | 10 | 20 | 30 | 40 | 50 | 60 | 70 | 80 | 90 | 100 |
| In-sample test | 87.5 | 85 | 84.5 | 84 | 82.5 | 83 | 81 | 80 | 77.5 | 75.5 | 75.5 |
| CRD | 46.1 | 51 | 50.6 | 52 | 51.8 | 52.3 | 52.3 | 51.8 | 50.2 | 49.8 | 49.8 |
| Amazon | 68.6 | 68.1 | 67.1 | 66.8 | 66.9 | 66.5 | 66.2 | 65.4 | 66.1 | 66.6 | 65.7 |
| Semeval | 56.7 | 56.6 | 56.3 | 56.4 | 56.2 | 56.4 | 56.4 | 56.2 | 56.8 | 56.4 | 56.4 |
| Yelp | 76.2 | 76.1 | 76 | 76.2 | 76.4 | 76.5 | 76.9 | 76.9 | 76.7 | 76.9 | 76.5 |
| | BiLSTM with Self Attention | | | | | | | | | | |
| In-sample test | 80.3 | 83.2 | 78.1 | 76.9 | 73.6 | 80.3 | 81.7 | 76.4 | 76.4 | 74 | 74.5 |
| CRD | 49.2 | 49.8 | 50.6 | 50.8 | 50.8 | 49.2 | 49.2 | 49.2 | 52 | 49.4 | 49.8 |
| Amazon | 50 | 49.8 | 50.5 | 49.8 | 50 | 49.7 | 49.7 | 50.1 | 50 | 50.3 | 49.8 |
| Semeval | 50 | 50 | 50 | 50 | 50 | 50 | 50 | 50 | 50 | 50 | 50 |
| Yelp | 50.5 | 50.4 | 50 | 50.5 | 50.8 | 50.3 | 50.1 | 50.9 | 50.7 | 50.8 | 50.5 |
| | Longformer | | | | | | | | | | |
| In-sample test | 97.5 | 98 | 98 | 97.5 | 97.5 | 97 | 97 | 97 | 96.5 | 94.3 | 92.8 |
| CRD | 93.4 | 93.4 | 93.9 | 94 | 92.4 | 91 | 92.2 | 91.7 | 90.2 | 86.6 | 74.5 |
| Amazon | 81.8 | 81 | 74.2 | 66.3 | 74.7 | 78.3 | 80.6 | 76.2 | 63.2 | 77.3 | 55.8 |
| Semeval | 80.3 | 79.9 | 69.4 | 64 | 73.4 | 77 | 78 | 74.6 | 60.6 | 78.3 | 56.4 |
| Yelp | 88.6 | 87.3 | 84.5 | 76.6 | 83.1 | 86.4 | 87.6 | 85.6 | 72.8 | 84.1 | 61.7 |
| Dataset | Percent noise in non-rationales | | | | | | | | | | |
| | SVM | | | | | | | | | | |
| In-sample test | 87.5 | 79.7 | 79.9 | 79.5 | 81.1 | 79.9 | 80.3 | 78.9 | 78.7 | 79.3 | 73.4 |
| CRD | 46.1 | 52.7 | 52 | 53.1 | 50 | 54.3 | 50.6 | 54.3 | 52 | 57.2 | 57.2 |
| Amazon | 68.6 | 68.1 | 66.2 | 67 | 65.8 | 68.8 | 65.3 | 64.4 | 65.3 | 63.1 | 61.9 |
| Semeval | 56.7 | 57.4 | 56.2 | 56.9 | 55.9 | 57.3 | 55.6 | 58.1 | 57 | 57.9 | 58 |
| Yelp | 76.2 | 76.4 | 75.4 | 76 | 75.6 | 75.8 | 74.2 | 74.3 | 73.6 | 73.7 | 71.5 |
| | BiLSTM with Self Attention | | | | | | | | | | |
| In-sample test | 80.3 | 76.9 | 80.8 | 79.3 | 78.8 | 77.9 | 76 | 76 | 63.5 | 73.6 | 66.8 |
| CRD | 49.2 | 50 | 50.2 | 50.4 | 50.2 | 50.8 | 51.4 | 47.9 | 48.8 | 47.5 | 48.4 |
| Amazon | 50 | 50 | 49.7 | 50.1 | 50.2 | 50.8 | 50.2 | 50 | 50.3 | 49.8 | 47.9 |
| Semeval | 50 | 50 | 50 | 50 | 50 | 50.1 | 50 | 50 | 50 | 50 | 50 |
| Yelp | 50.5 | 50.5 | 50.3 | 51.3 | 54.5 | 54.9 | 52.2 | 51.4 | 52.7 | 50.5 | 54.9 |
| | Longformer | | | | | | | | | | |
| In-sample test | 97.5 | 97.8 | 98 | 97.8 | 97.5 | 98.3 | 95 | 92.8 | 84.5 | 83.5 | 74.5 |
| CRD | 93.4 | 94.4 | 94.1 | 93.6 | 93.1 | 93.3 | 92.8 | 91 | 86.9 | 70.9 | 67.6 |
| Amazon | 81.8 | 80.9 | 75.9 | 75.8 | 79.7 | 68.9 | 81.4 | 72.4 | 71.2 | 63.5 | 55.2 |
| Semeval | 80.3 | 78.6 | 72.7 | 74.4 | 79.1 | 68.9 | 81.6 | 73.5 | 76.2 | 59.2 | 55.7 |
| Yelp | 88.6 | 88.1 | 84.1 | 84.8 | 87.5 | 81.3 | 89.3 | 82.2 | 82.2 | 70.3 | 61.5 |

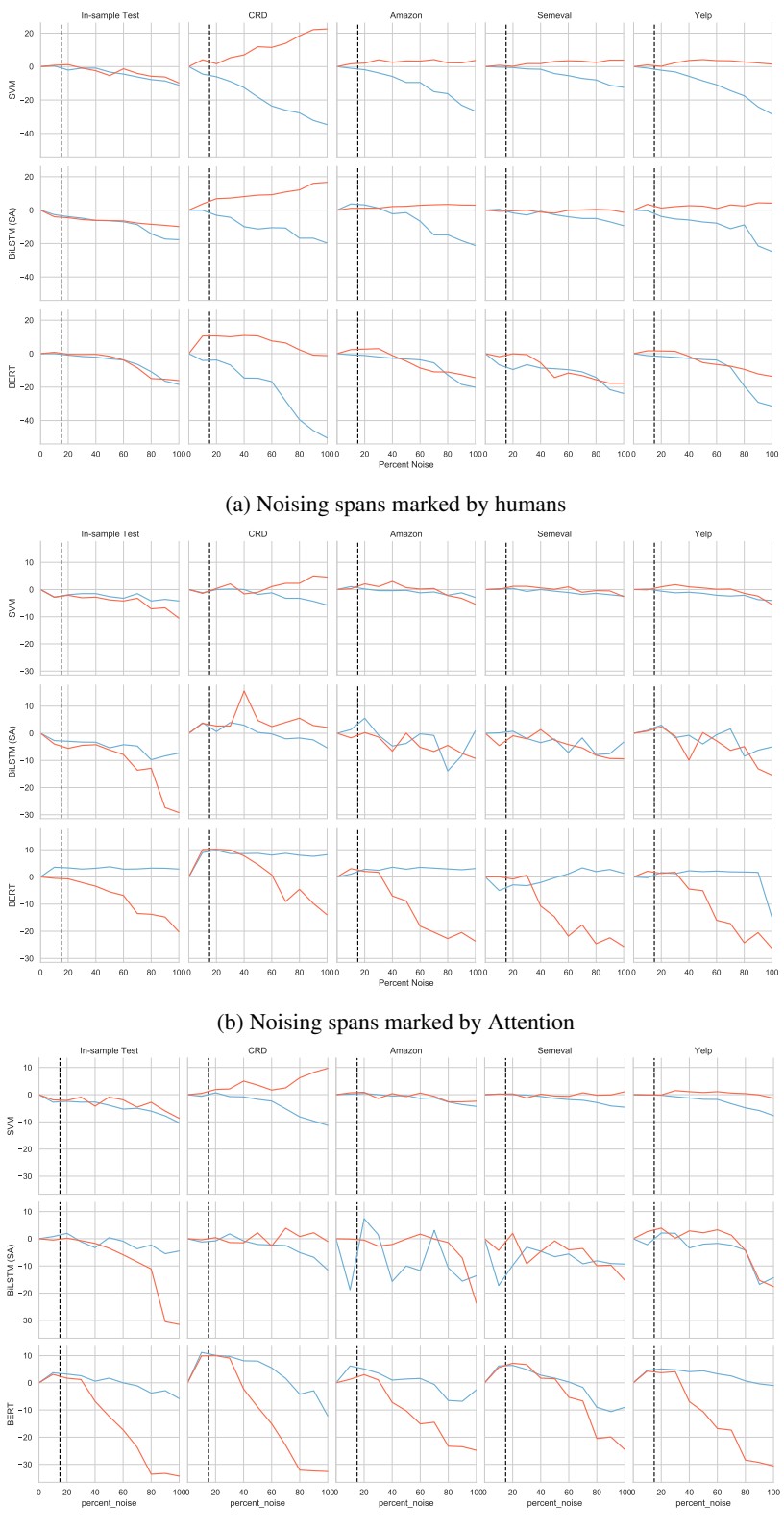

(a) Noising spans marked by humans

(b) Noising spans marked by Attention

(c) Noising spans marked by AllenNLP Saliency Interpreter

Figure 4: Change in classifier accuracy as noise is injected on *rationales* (in blue) or *non-rationales* (in red) for IMDb reviews from Kaushik et al. (2020). The vertical dashed line indicates the fraction of median length of *non-rationales* equal to the median length of *rationales*.

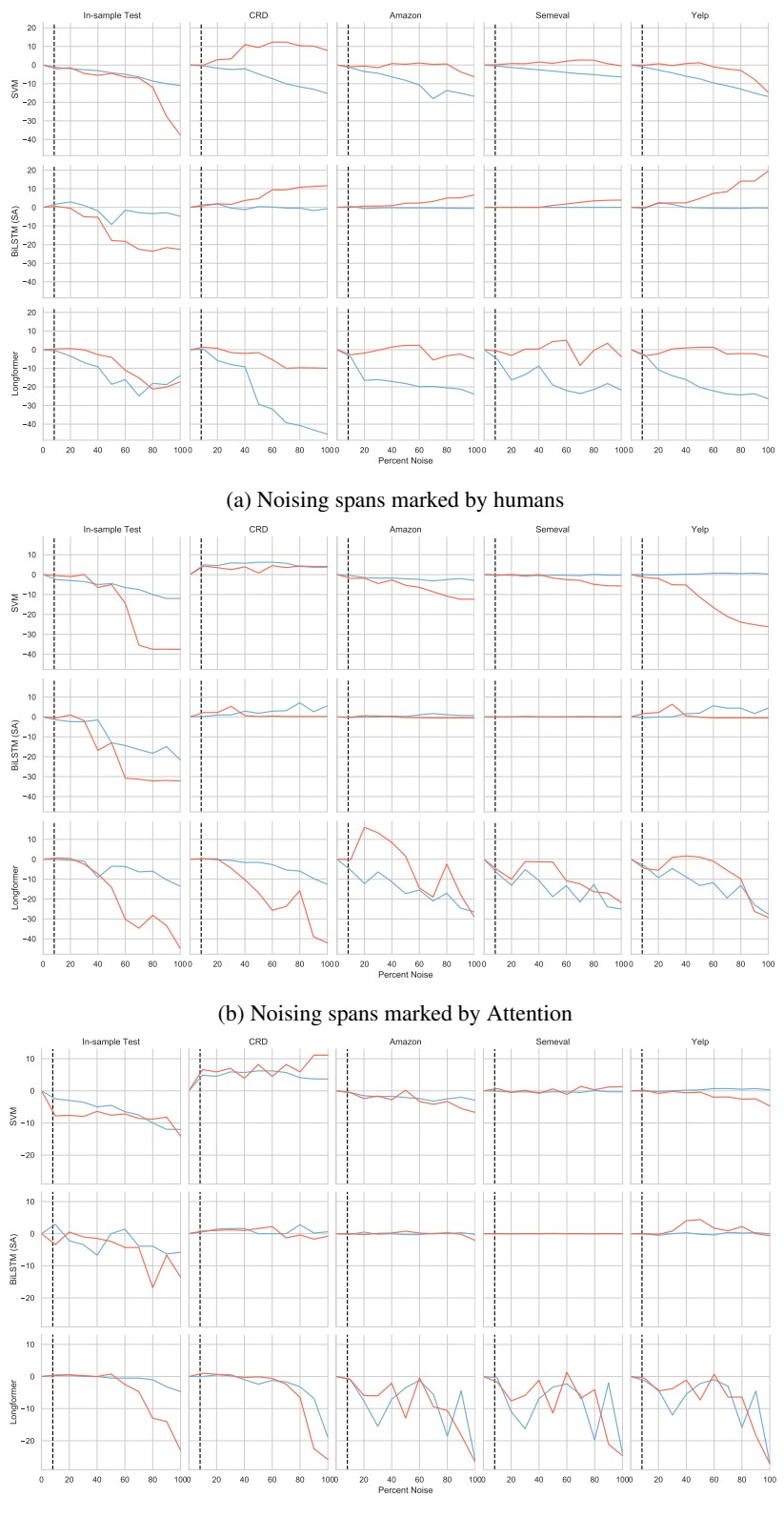

(a) Noising spans marked by humans

(b) Noising spans marked by Attention

(c) Noising spans marked by AllenNLP Saliency Interpreter

Figure 5: Change in classifier accuracy as noise is injected on *rationales* (in blue) or *non-rationales* (in red) for IMDb reviews from Zaidan et al. (2007). The vertical dashed line indicates the fraction of median length of *non-rationales* equal to the median length of *rationales*.

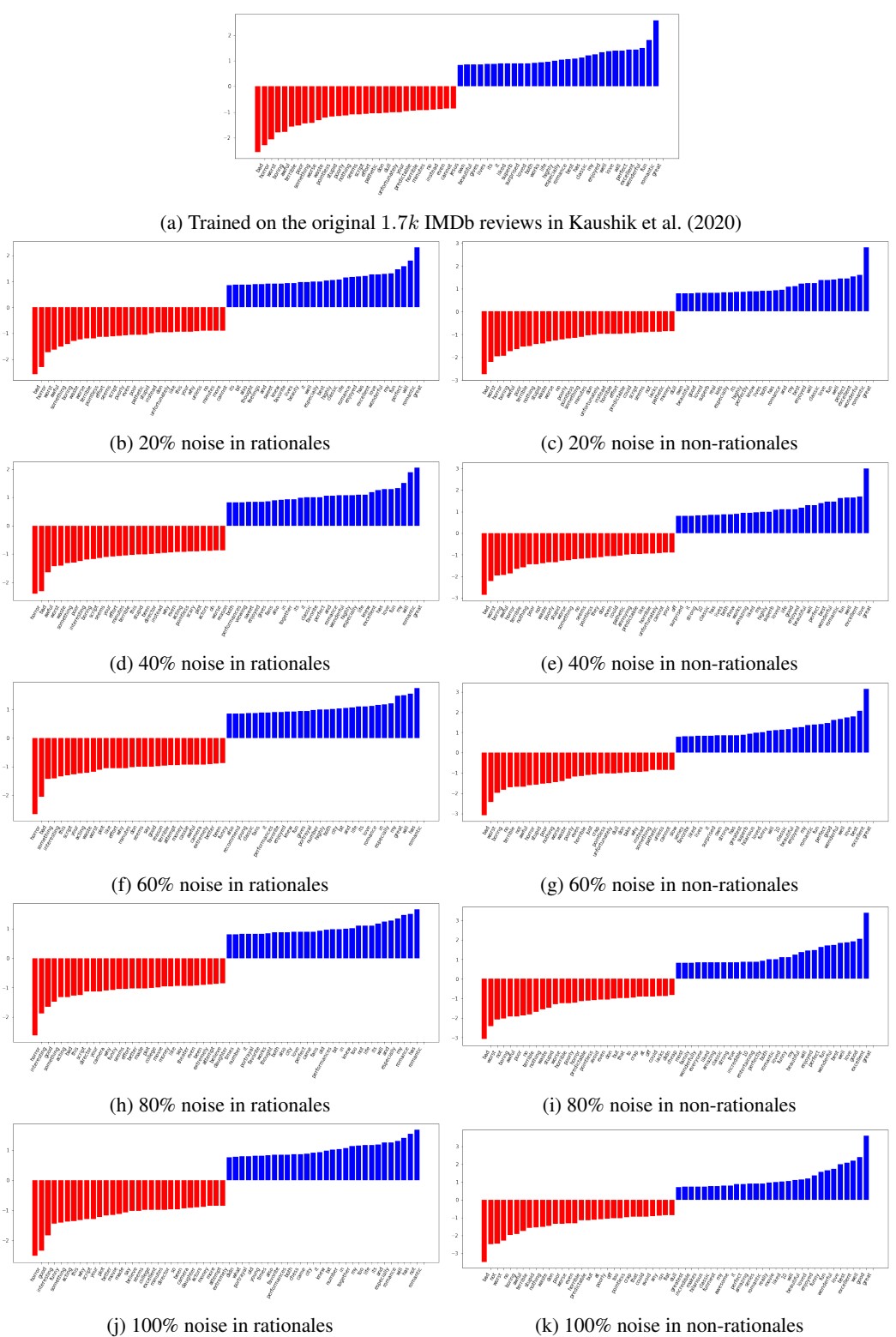

Figure 6: Most important features learned by an SVM classifier trained on TF-IDF bag of words. Rationales are identified by humans.

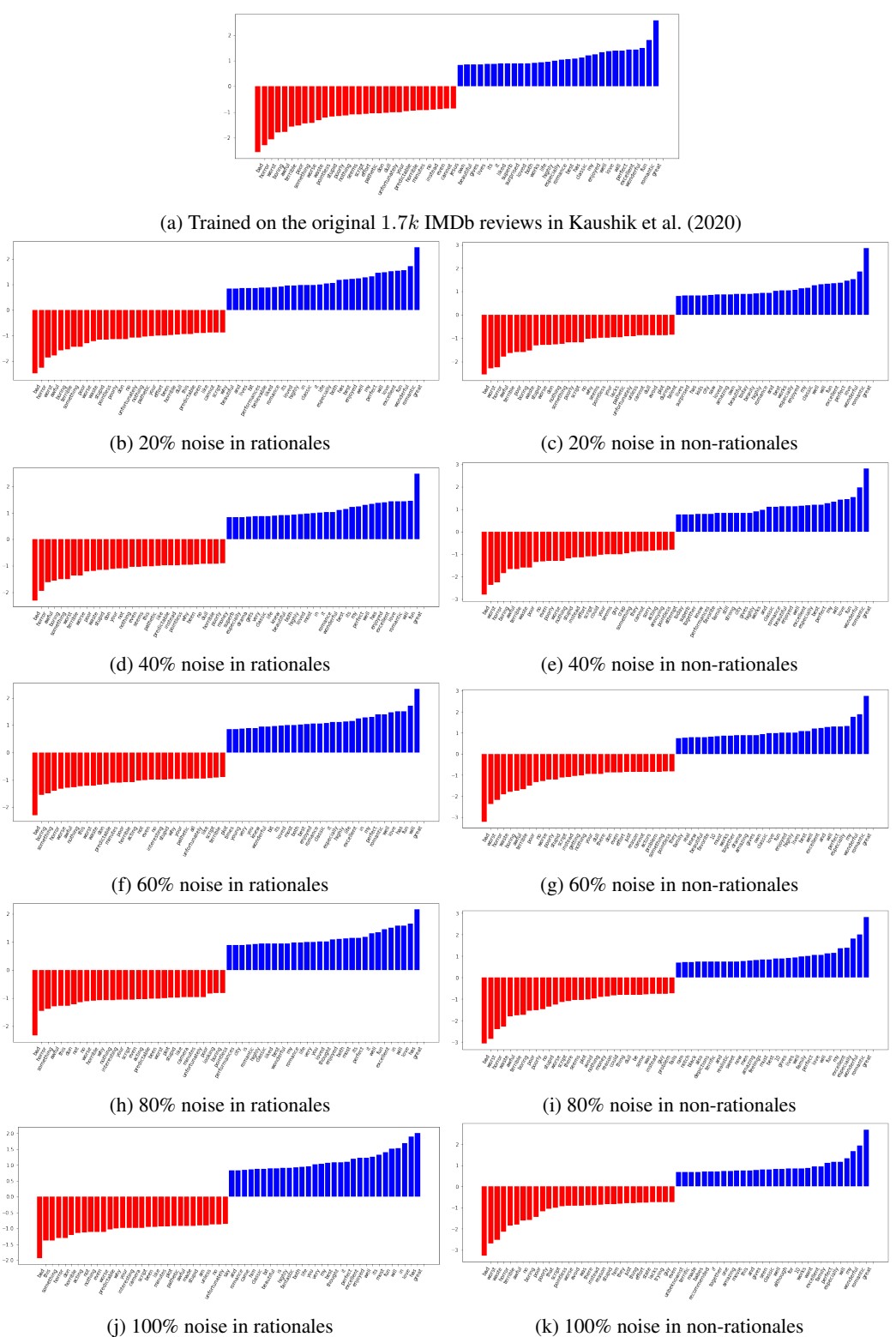

Figure 7: Most important features learned by an SVM classifier trained on TF-IDF bag of words. Rationales are identified as tokens attended upon by a BiLSTM with Self Attention model.

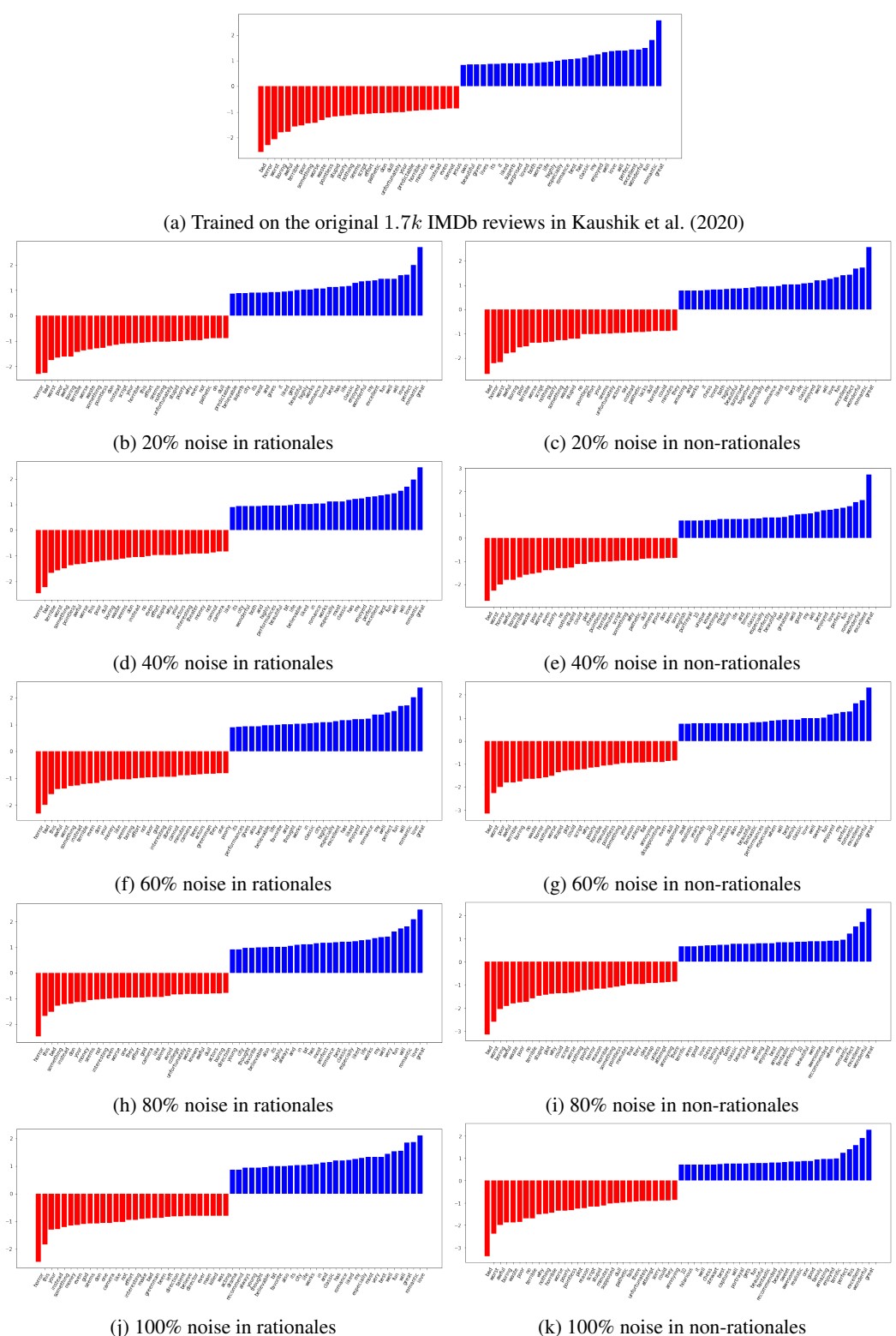

(a) Trained on the original $1.7k$ IMDb reviews in Kaushik et al. (2020)

(b) 20% noise in rationales

(c) 20% noise in non-rationales

(d) 40% noise in rationales

(e) 40% noise in non-rationales

(f) 60% noise in rationales

(g) 60% noise in non-rationales

(h) 80% noise in rationales

(i) 80% noise in non-rationales

(j) 100% noise in rationales

(k) 100% noise in non-rationales

Figure 8: Most important features learned by an SVM classifier trained on TF-IDF bag of words. Rationales are identified as tokens marked by the AllenNLP Saliency Interpreter.

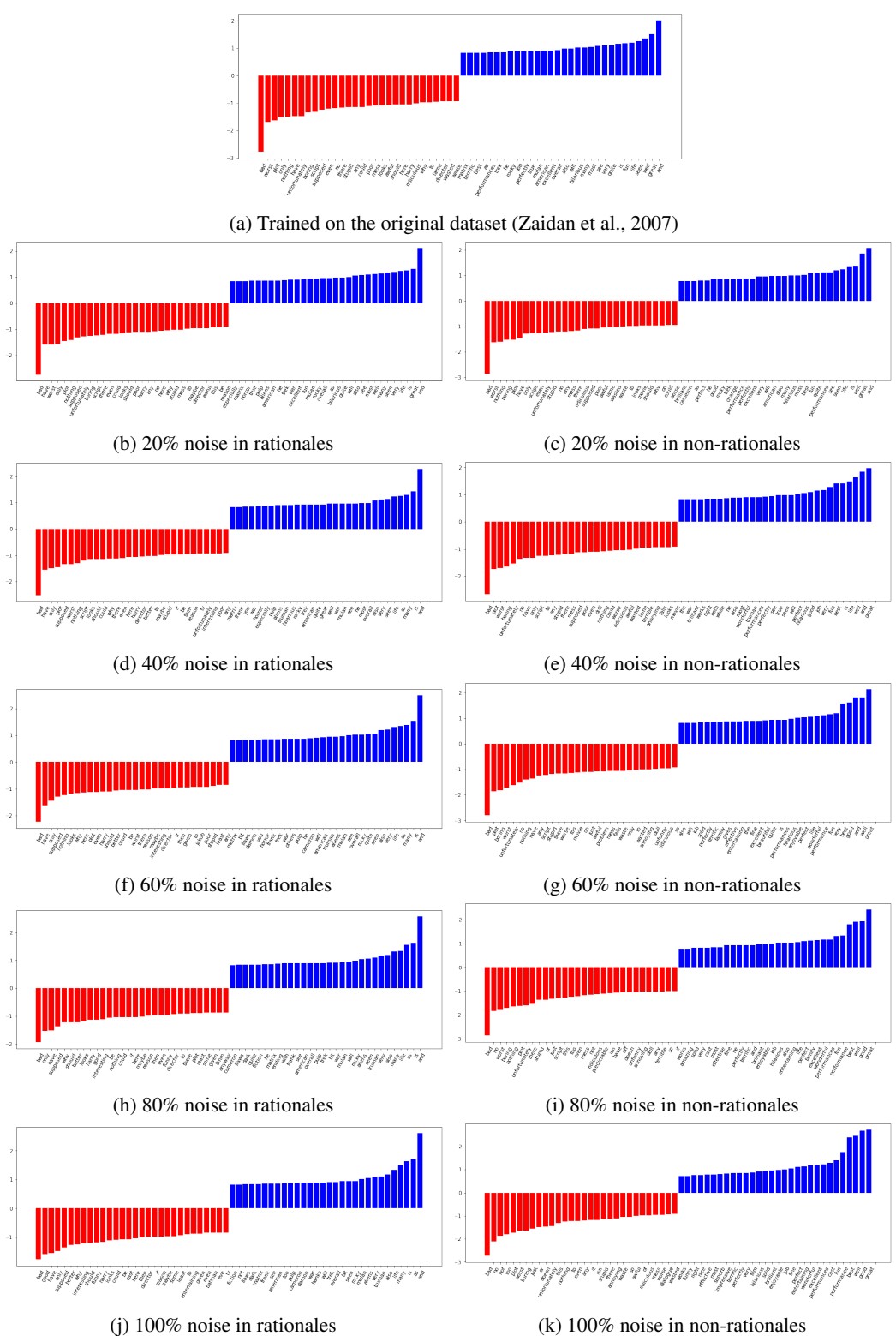

Figure 9: Most important features learned by an SVM classifier trained on TF-IDF bag of words. All noise inserted on rationales identified by humans.

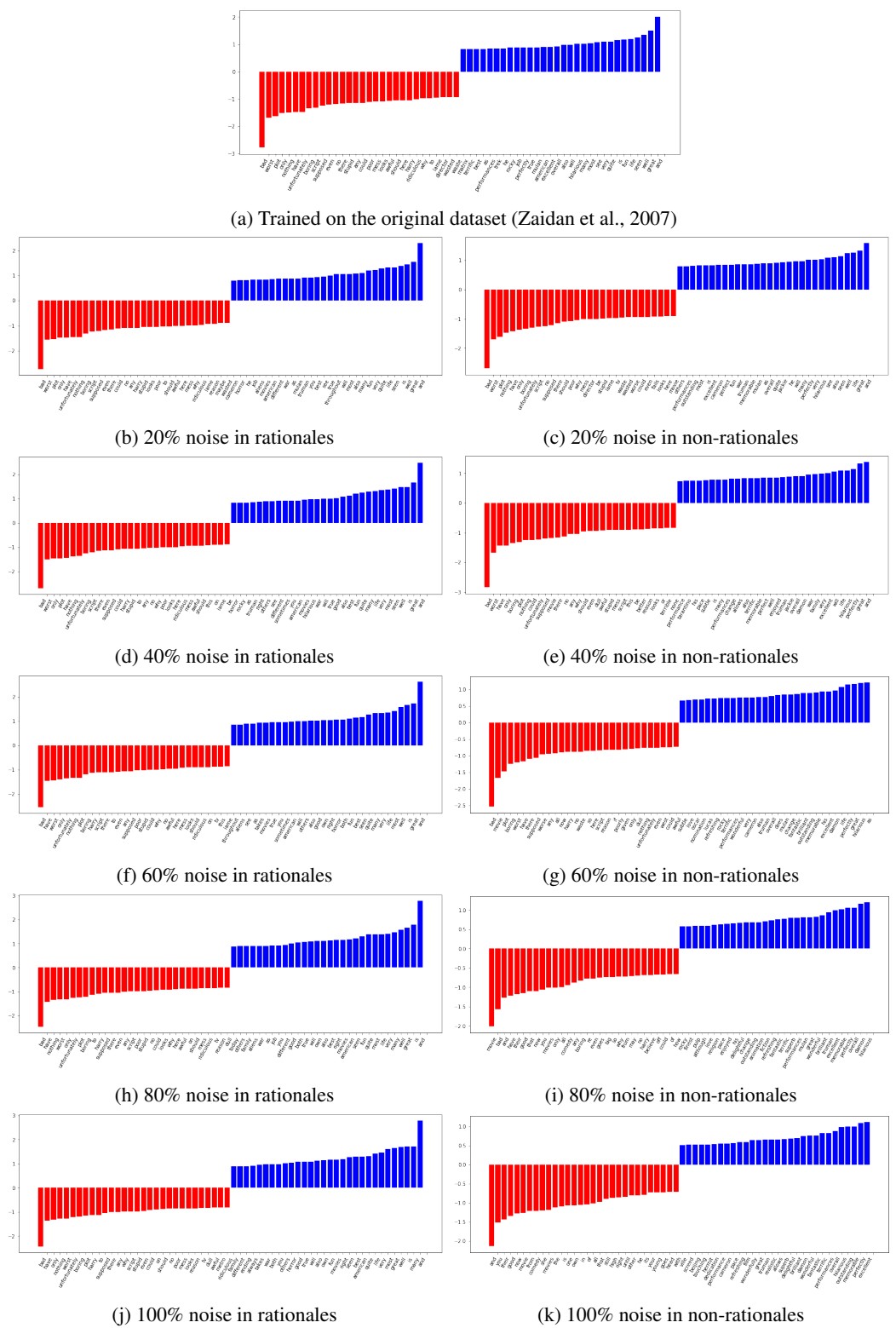

Figure 10: Most important features learned by an SVM classifier trained on TF-IDF bag of words. Rationales are identified as tokens attended upon by a BiLSTM with Self Attention model.

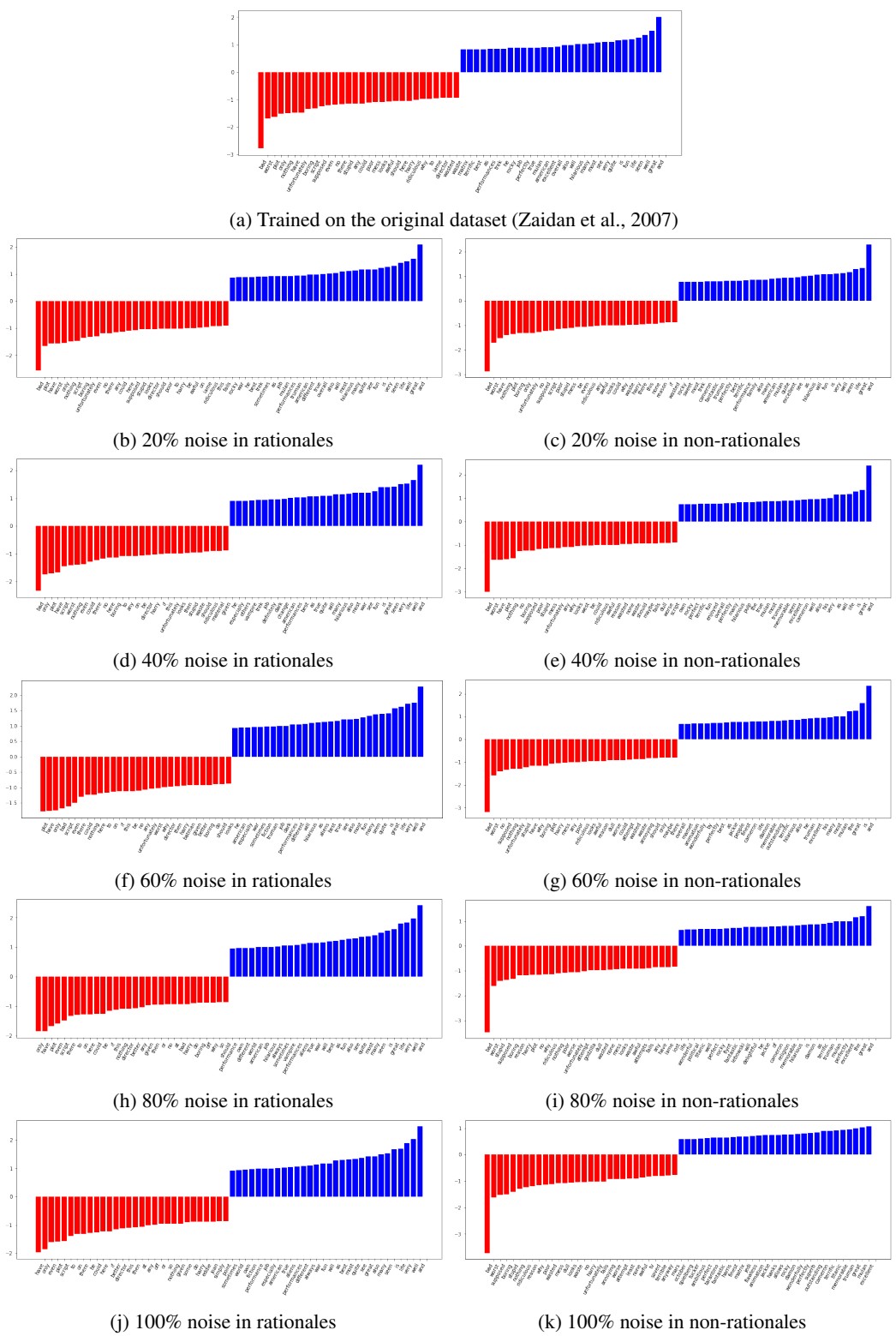

Figure 11: Most important features learned by an SVM classifier trained on TF-IDF bag of words. Rationales are identified as tokens marked by the AllenNLP Saliency Interpreter.

Table 9: Accuracy of various models for sentiment analysis trained with various datasets. O refers to the in-sample test set from Kaushik et al. (2020) whereas R refers to the counterfactually revised counterparts of the same.

| Training data | SVM | | NB | | BiLSTM w/ SA | | BERT | |
|---|---|---|---|---|---|---|---|---|
| | O | R | O | R | O | R | O | R |
| Orig. (1.7*k*) | **80.0** | 51.0 | **74.9** | 47.3 | **78.0** | 49.4 | **87.4** | 82.2 |
| CRD (1.7*k*) | 58.3 | **91.2** | 50.9 | **88.7** | 63.8 | **82.0** | 80.4 | **90.8** |
| Hu et al. (2017) | 56.3 | 68.0 | 57.1 | 71.1 | 55.1 | 67.0 | 66.3 | 74.4 |
| Li et al. (2018) | 41.3 | 54.1 | 37.8 | 58.0 | 49.9 | 49.4 | 37.9 | 55.9 |
| Sudhakar et al. (2019) | 47.1 | 55.9 | 42.6 | 58.8 | 49.9 | 49.4 | 43.7 | 51.8 |
| Madaan et al. (2020) | 61.2 | 77.3 | 50.2 | 75.2 | 59.3 | 69.7 | 70.6 | 81.6 |
| CAD (3.4*k*) | 83.7 | **87.3** | 86.1 | **91.2** | 80.3 | **84.8** | 88.5 | **95.1** |
| Orig. & Hu et al. (3.4*k*) | 82.1 | 66.4 | 81.5 | 55.1 | 76.4 | 63.9 | 87.1 | 89.3 |
| Orig. & Li et al. (3.4*k*) | 73.3 | 55.7 | 77.9 | 53.3 | 69.5 | 53.3 | 80.3 | 79.5 |
| Orig. & Sudhakar et al. (3.4*k*) | 74.1 | 56.1 | 79.1 | 51.4 | 71.4 | 55.7 | 89.1 | 90.8 |
| Orig. & Madaan et al. (3.4*k*) | 83.8 | 65.4 | 82.1 | 67.4 | 75.5 | 64.1 | 83.5 | 81.6 |
| Orig. (3.4*k*) | **85.1** | 54.3 | 82.4 | 48.2 | **80.1** | 57.0 | **90.2** | 86.1 |

Table 10: Accuracy of BERT trained on subsample of SNLI (DeYoung et al., 2020) (where number of rationale tokens and non rationale tokens are within 30% of one another) as noise is injected on human identified *rationales/non-rationales*. RP and RH are Revised Premise and Revised Hypothesis test sets in Kaushik et al. (2020). MNLI-M and MNLI-MM are MNLI (Williams et al., 2018) dev sets.

| | Percent noise added to train data rationales | | | | | | | | | | |
|---|---|---|---|---|---|---|---|---|---|---|---|
| Dataset | 0 | 10 | 20 | 30 | 40 | 50 | 60 | 70 | 80 | 90 | 100 |
| In-sample test | 90 | 87.1 | 83.5 | 80.3 | 80.8 | 78.9 | 77.8 | 77.5 | 73.5 | 67.9 | 69.7 |
| RP | 66.9 | 66.4 | 60.4 | 57.9 | 56.9 | 54.4 | 52.3 | 51.3 | 51.4 | 51.2 | 51.5 |
| RH | 79.2 | 75 | 69.8 | 67 | 66.5 | 64.2 | 63.5 | 65.7 | 64.9 | 61.7 | 61.8 |
| MNLI-M | 74.1 | 66.4 | 61.9 | 59.8 | 59.4 | 57.4 | 54.5 | 56.6 | 55.7 | 54.7 | 54.6 |
| MNLI-MM | 76.1 | 66.5 | 61.4 | 59 | 58.5 | 56.5 | 53.6 | 56 | 55.6 | 54.2 | 54.4 |
| | Percent noise added to train data non-rationales | | | | | | | | | | |
| Dataset | 0 | 10 | 20 | 30 | 40 | 50 | 60 | 70 | 80 | 90 | 100 |
| In-sample test | 90 | 88.7 | 87.5 | 85.2 | 85.7 | 84.2 | 83.4 | 82.2 | 79.3 | 77.2 | 74.9 |
| RP | 66.9 | 68.3 | 66.2 | 62.7 | 64.5 | 63.3 | 62.6 | 61.7 | 61.5 | 61.5 | 62.5 |
| RH | 79.2 | 78.4 | 77.4 | 75.9 | 74.6 | 73.4 | 72.8 | 72.2 | 73 | 70.6 | 70.8 |
| MNLI-M | 74.1 | 67.6 | 66.5 | 64.4 | 65.4 | 62.8 | 62.6 | 62.1 | 61.9 | 61.8 | 61.9 |
| MNLI-MM | 76.1 | 68 | 67.6 | 65.1 | 65.1 | 63.1 | 63.1 | 62.3 | 61.6 | 61.3 | 61.1 |

