# OpenReview forum: "Explaining the Efficacy of Counterfactually Augmented Data"
_ICLR.cc/2021/Conference — ICLR 2021 Poster_

### Official Review · AnonReviewer2 · 2020-10-21
**Spurious correlations no longer exist in counterfactually augmented data**

**Rating:** 8
**Confidence:** 4

**Review:**

This paper proposes an analysis of the benefits when using counterfactually augmented data (CAD) for classification purposes. The proposed framework is causal inference and DAGs. They focus on the problem of spurious correlations and how they affect out-domain generalization.

They study a simple scenario with linear models and show that in this scenario, when using CAD,  spurious correlations are no longer used by classifiers. These qualitative insights as well as framing the problem in a causal model are the main contribution of the paper. This is not a minor contribution: up to now the discussion about the benefits of CAD where only supported by empirical findings. Framing the problem, even if considering simplified models, is a critical step.

I appreciate the dual causal model that is studied (causal and "anticausal"), not limiting the analysis to the most common interpretation. They show that both models mean to the same conclusions.

The paper will be more informative with a discussion about the noise model in the case of the noisy proxy (pg. 4). Is this the most probable noise model? Are there alternatives?

The paper presents a large series of experiments to support the proposed hypothesis. Experiments are well designed and results are convincing.

The paper opens a full line of research, especially if we consider the generalization of these results to other related domains.

---

> ### Author Response · Authors · 2020-11-24
> **Reply to R2**
>
> We thank you for positive feedback and for championing our paper. We are especially glad that you recognize the value of providing a deeper conceptual framing for CAD and see it as a critical step towards advancing this emerging research area.
>
> Based on your suggestion, we have amended the paper, providing a more informative discussion about the noisy proxy model and why we believe it is a useful abstraction for thinking about what happens in practice (Section 3).

---

### Official Review · AnonReviewer4 · 2020-10-28
**Nice paper**

**Rating:** 7
**Confidence:** 3

**Review:**

This paper presents a perspective on counterfactually augmented data (CAD) and experiments providing evidence that CAD elicited from humans indeed bears resemblance to interventions on labels or variables that are causally related to the labels.

This is already the stated intuition behind CAD and why CAD may help with learning a model that generalizes well out of domain. However, as this paper argues, questions remain over the exact mechanism of CAD’s benefits, and whether, for example, existing automated methods (such as attention and style transfer) may be used to accomplish the same goal as human interventions.

The main contribution of the paper is new indirect evidence in favor of the original intuition behind CAD. This is made possible through an observation provided over a toy model of Gaussian structural causal models. Section 3 shows that noise in the causal variable increases a learned model’s dependence on a confound, whereas noise in a confound does not affect the model in the absence of noise on the causal variable. More weight given to the confound will hurt out-of-domain generalization by the model. All this being the case, we would predict that corrupting causal inputs in training data would hurt out-of-domain performance, while corrupting non-causal inputs would help it (even if it hurts training).

This provides a framework for the experiments, which show this is indeed the case for certain human-provided annotations of rationales, providing corroborating evidence that human-annotated rationales or counterfactual augmentation do correspond to causal variables. Automated methods such as learned attention or style transfer, on the other hand, do not exhibit this trend, providing evidence that they do not pick out the appropriate causal variables.

### Strengths

The paper has a fairly clear logical flow and provides clear theoretical and empirical components which are connected well. I also think the motivation is very nicely done. Lots of relevant work is brought together and a good case is made for the importance of the questions addressed by this paper.

### Weaknesses

I don’t have any serious complaints. The contribution is a tad narrow, but it makes progress on some tricky and difficult questions. The experiments also only produce *corroborating* evidence of CAD’s status as implicating causal variables, and we already know by construction that there is a causal aspect to these perturbations, so it’s not exactly an Earth-shaking result. But the experimental framework gives what seems to be pretty good evidence that other automated methods *don’t* implicate causal variables, so that’s nice. The main value seems to be in the theoretical observations about the effect of noise on out-of-domain performance and what that tells us about causality. These observations in the paper seem fairly straightforward, but I’m not aware of any literature making the same point. (However, I am not perfectly familiar with the literature and might have missed it.) Regardless, the application of these observations to experiments is interesting and provides useful conceptual scaffolding for future research about causal variables in models (such as deep NNs) without such explicit notions.

### Recommendation

Accept. The paper seems sound, is well written, and addresses an important problem. The contribution may not be huge but seems to me worth publishing.

### More comments/questions

The bolded paragraph on P. 5 might be a bit much. It makes a particular claim about “language meaning” which implicitly views meaning as corresponding to the causal connection between input language and output labels. The notion of language “meaning” is nuanced and it’s not clear whether this claim is true for all tasks, where labels may be annotated based on broader (or indeed narrower) inferences regarding the generative process of the text. Since this isn’t purporting to be a paper about language meaning, I would suggest staying away from this.

### Typos, style, etc.

* P. 2: I think you should be able to render (Wright et al., 1934; Figure 1) more naturally by using the [bracketed arguments] in \citep. ...Not sure how this plays with hyperref.
* P. 3: period before “and unintentional”
* P. 6: “use SVM” -> “we use SVM”

---

> ### Author Response · Authors · 2020-11-24
> **Reply to R4**
>
> Thank you for the thoughtful review and positive assessment. We are glad you think that the paper is well motivated, has a clear logical flow, and the application of our observations provide useful conceptual scaffolding for future research about causal variables in ML models.
>
> Per your suggestion, we have toned down our discussion on the stability of causal relations (Section 3) to avoid inviting unnecessary debates about “meaning” in language.
>
> Thank you also for catching these typos, we have made all recommended fixes in the updated version.

---

### Official Review · AnonReviewer3 · 2020-10-29
**Very exciting idea but needs to be developed more before publishing**

**Rating:** 6
**Confidence:** 3

**Review:**

Update after author response:

I appreciate the authors' efforts to address my concerns. I still find the paper's insights lacking in the critical middle ground between theory and practice, but understand that drawing these connections is a long game. I believe the paper leads research in an important direction and so can be published despite these flaws.

Summary:

The paper investigates the issue of counterfactual data augmentation (CDA), i.e., the process by which humans augment a model's training data by making minimal edits necessary in order to flip the label. For example, in the case of sentiment analysis, humans might edit a positive movie review in order to make it negative, changing as few words as possible. The goal of CDA has been to reduce models' reliance on spurious features (e.g., gender bias, dataset artifacts) and instead encourage the model to base its predictions on features which will generalize outside of the training data. This paper makes two contributions.
* First, the authors attempt to analyze CDA through the lens of causal inference, looking at a toy setting of Gaussian OLS. Specifically, they consider a simple graph containing a causal feature and a non-causal feature, where these features may be observed with some noise. They walk through a simple derivation for the slopes of the OLS in terms of the variance on these features and show that the slope on the causal feature will decrease to 0 as noise on the causal feature increases, and that the slope on the non-causal feature will increase. The takeaway is that noising non-causal features should lead a model to rely more on causal features (and thus generalize better), whereas noising the causal features should lead the model to rely on spurious features and thus perform worse overall.
* Second, equipt with the above takeaway, the authors perform a series of empirical studies in which they add noise to allegedly causal vs. non-causal spans of text and evaluate whether models generalize better as a result. Specifically, they focus on some recently released CDA datasets that take the form of e.g., movie reviews in which spans have been marked by humans to denote which words are most important for predicting the label. The authors show that noising these spans leads models to perform worse while noising random other spans causes models to perform better. The experiments are also run using attention (as opposed to human annotation) in order to identify "causal" spans, and using automatic style transfer techniques to perform the CDA, though the results of these automatic methods suggest that they perform less well than using the human-identified spans.

Overall, the paper has potential to make an important contribution, but I feel it falls short in its current form. The idea of analyzing CDA in terms of causal graphs is exciting, but its not fully fleshed out here. I don't fault the authors for beginning with a simple/toy setting, but I do think they owe the readers more discussion of the implications and limitations of this setting (see my questions below) in order for the theoretical results to be useful. I also found the tie between the empirical results and the theoretical ones to be tenuous--the empirical results are unsurprising (they can be paraphrased as "adding noise to important features causes models to perform worse") and could probably be explained by a number of theoretical frameworks. Thus, its not clear to me why we should accept them as evidence of the causal explanation (I elaborate below). Finally, I think the paper is a bit hard to read and lacking in experimental details (more comments below). For example, the results that use automatic style transfer are a bit of an afterthought, and so its hard to know what to take away from Table 2.

Strengths:
* Weighs in on a very important open question: how can we reason rigorously about counterfactual data augmentation?
* Makes efforts toward both a theoretical and an empirical contribution

Weaknesses:
* Discussion surrounding theoretical contribution is lacking, making application appear quite limited
* Connection between theory and empirical results is tenuous, and authors should discuss alternative explanations for why we'd see these empirical trends
* Paper is hard to follow. The empirical results try to do too much, and so no one experiment is properly explained.

Detailed Questions/Comments:
* Better discussion of theoretical assumptions: You focus on a simple Gaussian OLS model for your theoretical discussion, which you acknowledge is a toy setting and I appreciate that. I don't think there is any harm in these toy settings, but I do think that there should be a more informed discussion about exactly how much we can/should takeaway from it. For example, these causal graphs assume that training amounts to learning a relative weighting on features, when in fact modern NLP models are simultaneously learning which features to extract and how to weight them, which means many things other than noise might mediate the models' reliance on x1 vs. x2. For example this paper (https://arxiv.org/pdf/2004.15012.pdf) suggests that features differ in "accessibility" and that might influence which feature is used, regardless of noise. I think the final prediction you make (last paragraph before Section 4) which informs your empirical results is sensitive to these assumptions and thus it requires discussion. Again, I realize you own the speculative nature of these claims and I appreciate that, but even so, that doesn't remove the obligation of connecting the dots. For example, can you comment on: If your assumptions are not met, what does that mean? How should we interpret your empirical results in that case? Are the findings rendered completely irrelevant, or can we take some relaxed interpretation of them? I think a good theoretical contribution requires some discussion along these lines.
* Alternative explanations for empirical trends: Your primary empirical results show that noising unimportant features hurts models less than noising important features. That result alone seems unsurprising. What is interesting in your paper is that these results are consistent with a prediction that was made using your causal analysis, and thus are interpreted as evidence consistent with the causal model. I think that some additional discussion is warranted, since there are many (arguably simpler) models other than the causal one that may also generate these data. Its possible I missed something in your proofs, so correct me if I am wrong, but my understanding is that the same data could be explained purely in terms of which feature is most strongly correlated with the label during training. I.e., by design, the causal feature is more strongly correlated with the label than the spurious feature, and as you add noise, the model just ends up relying on whichever feature is more strongly correlated. This would be the case regardless of the causal graph. Is that a fair paraphrase? If so, its hard to say that these data substantiate anything about the causal nature of the features in play in CDA. But please correct me if I am wrong, its likely I missed a nuance of your argument.
* Hard to follow: Overall, I'd recommend making fewer points and spending more time communicating the important ones. I found Tables 1 and 2 hard to follow given how little detail was provided in the text. In Figures 2 and 3 (arguably the main results) its really hard to see what the main takeaway is. You might consider putting rationales on the same plot as nonrationales so that we can clearly see when and by how much they pull away from one another. Other little details are missing as well (e.g., what is lambda_c in eq. 4? this was never defined). I'd recommend asking someone unfamiliar with the work to give it a read through and nit pick for clarity.

---

> ### Author Response · Authors · 2020-11-24
> **Reply to R3**
>
> Thanks for the detailed, critical review. We are glad that you believe this to be a very exciting idea, and take your criticisms seriously. However, we believe that a few major points owe more to misunderstandings that could be addressed via this rebuttal and our improvements to the exposition.
>
> **Re: More discussion of limitations** We agree that this transparently communicating limitations is fundamental to the usefulness of theory and have revised the draft accordingly (Section 3).
>
> **Re: the empirical results are unsurprising (they can be paraphrased as "adding noise to important features causes models to perform worse") and could probably be explained by a number of theoretical frameworks.**
>
> We disagree with the reviewer on two key counts:
> (i) This is not an accurate characterization of our findings. We are not simply saying that noise on important features makes models perform worse. Rather, we conjecture that noise on causal features causes significantly greater degradation out of domain than in domain, and show that this indeed holds on real world datasets. Moreover our experiments show that this holds for the *causal features* as a identified by counterfactually augmented data and by human-annotated rationales, but not that the same patterns do not hold for the *important features* assessed by automated feature importance methods, including attention weights, and (added in the updated draft) gradient-based saliency methods.
>
> (ii) It’s important to note that our theory preceded our experiments. While it’s easy to conjure numerous theories compatible with retrospective findings, we first posited a theory and then found qualitative evidence that bear out its predictions. We believe that this is a key measure of the value of a scientific theory: that it predicts phenomena that can subsequently be experimentally verified.
>
> **Re: hard to read; clarity on experimental details** Thanks for this critical feedback. We have made an aggressive revision to improve the clarity of the draft and hope that you find the narrative and experiments better presented.
>
> **Re: assumptions and insights absent faithfulness** We agree more can be done to help connect the dots and have attempted to do this throughout the updated draft both by fleshing out some arguments and by explicitly recognizing limitations. Indeed many assumptions in the linear model are not met in practice. However such stylized models are often nevertheless useful, with utility assessed, in part, by the ability of the theory to provide qualitative insights that are born out in practice. The fact that our findings (derived from experiments inspired by our toy model) align with the theory’s predictions is itself evidence of the theory’s utility. We discuss this in Section 3 of our updated draft.
>
> **Re: alternative explanations** We are grateful for your willingness to entertain clarifications/counterarguments and are happy to provide one:
>
> It’s important to clarify what is meant by “important” and what is meant by “hurt”. Our findings speak to the impact of *causal features* (but not salient non-causal features) on out-of-domain (but not in-domain) performance. Your assertion that the same phenomena could be explained in purely associative terms is contradicted by our experiments showing that “important features” assessed by the most popular purely associative feature attribution methods do not confirm the same pattern.
>
> **Re: hard to follow** We have taken care to improve the exposition, especially around communicating the experimental results.
>
> Per your suggestion we have overlaid the rationale and non-rationale results on the same plot and placed them in the Appendix (Figures 4 and 5) for your consideration.

---

### Official Review · AnonReviewer1 · 2020-11-01
**Interesting study of the effect of causal vs non-causal features on out-of-domain generalization**

**Rating:** 7
**Confidence:** 4

**Review:**

# Edit after author response
I thank the authors for their detailed response and for taking into account many of my comments. One issue that remains is the disconnect between toy and natural scenarios. There is some added discussion, which is helpful, but actually sketching out and conducting experiments, including intermediary steps moving from a full toy example to the naturalistic case, could be very helpful here.


# Summary
This paper studies the impact of counter-factually augmented data on out-of-domain generalisation. The paper starts with a toy example of a structural causal model with Guassian linear models, where noise is added to causal or non-causal features. Increasing noise on casual features affects the least-squares estimates, while increasing noise on non-causal features does not. The paper draws an analogy between this scenario and counter-factual text editing, where edited spans (rationales) are presumed to be the causal features. A hypothesis is put forth that adding noise to rationales (causal features) would cause a model to rely on non-rationales (non-causal features) and result in poorer out-sample performance, while adding noise to non-rationales would lead to worse in-sample performance but better out-sample performance. Experiments on sentiment and natural language inference (NLI) datasets mostly confirm this hypothesis, with some exceptions to be discussed. The experiments include three ways to identify rationales (human edits, human-identified spans, and spans identified by self-attention). Models are trained with or without noise (random tokens replacing real tokens) on rationales or non-rationales.

# Evaluation
1. The paper studies an important question of how models may rely on causal and non-causal features, and how this impacts out-of-domain generalisation. The toy example is useful for proving the case in a simple setup and getting intuition. The experiments with NLP datasets are comprehensive.
2. That said, I think the emphasis on counterfactually augmented data is not conducive to the arguments and experiments in the paper. I think the paper may better be pitched as a study of causal vs non-causal features and the impact of them on out-of-sample generalization. The counter-factual data augmentation (text editing) is one way that the paper uses to identify such features, but the paper in fact has other ways to do so. But
3. There is a potential disconnect between the toy example and the NLP experiments, as acknowledged in the paper. I don't have a good suggestion now on how to bridge this gap, but perhaps there can be intermediary steps moving from a full toy example to the naturalistic case.
4. There are a lot of experiments and results, but the presentation and discussion of them are hard to follow. There is one long paragraph that discusses the results on page 7. On the one hand, it is full of details and can be split into several paragraphs. On the other hand, there are missing discussions of various aspects, which I discuss below. I would split these paragraphs in several, and discuss each aspect of the experiment in more detail. See more on this below.
5. Conversely, the introduction is pretty long and perhaps can be shorten to allow for more space. I also give suggestions on what to cut out below.

I'd be willing to reconsider my evaluation if these issues are addressed.


# Main comments

## Introduction
1. "When the causal features are subject to observation noise, the non-causal features become salient" - what does "salient" mean here?
2. Counterfactual data augmentation (CAD) is formalized as intervention on causal features (d-separation of label from non-causal features). Alternatively, CAD is formalized in anti-causal framework. "Here it is us who intervene on the label and the role of the editors is that of the structural equation." What does this mean? In general, mode background on the anti-causal framework may be helpful.


## Toy example
3. Why is there noise in x2? (eq 3) Differentiate between the noise in the data generation (eq 3 terms) and the measurement noise mentioned right after.
4. What is meant by "intervention" (for example on x1) in this case?

## Empirical study
5. There are many aspects that may impact the behaviour: how rationales are identified (human-edited, human-annotated, automatically attributed), whether noise is added to rationales or to non-rationales, the test domain (in-sample / out-sample, but also various out-sample datasets), and which model is used (SVM, LSTM w/SA, BERT, etc.).  It looks like all the results are available, but more careful discussion of different factors would be helpful. For example, how results differ by out-sample dataset is not discussed, and how model choice affects the trends is only briefly hinted at in the conclusion.
6. In NLI, when noise is added to rationales, out-sample performance drops more than in-sample performance, while when noise is added to non-rationales, in-sample performance drops more than out-sample performance. This is consistent with the sentiment case. However, the drop in in-sample accuracy when adding noise to non-rationales is smaller than the drop when adding noise to rationales. This is different from (some?) cases in the sentiment experiment, where adding noise to non-rationales led to a larger drop in in-sample accuracy compared to adding noise to rationales. Why is there a difference between NLI and sentiment? It seems like the NLI models may not rely as much on non-rationales, so adding noise to them doesn't impact in-sample performance much.
7. The discussion of the case of automatically-identified rationales is very brief and I think more can be said there about the results. There is also some potential concern with using attention for feature attribution, and using a gradient-based feature attribution method might be better.
8. One major difference from the toy example and the hypothesis made in the first part of the paper is that adding noise to non-rationales does not always result in better out-sample performance. Sometimes it results in a more moderate part. Presumably, this can be explained by rationales not being perfectly aligned with causal features, or by models being able to ignore some of the noise, or some other reason. I think it's an important point to discuss.
9. The style transfer experiments seem disconnected from the main theme of the paper (what are the causal/non-causal features? What is the noise? etc.) They not properly discussed and contrasted with other experiments. It might be better to leave those out completely.
10. Layout: figures and tables do not appear next to where they are discussed in text, making it hard to navigate the paper. Some tables are mentioned in the main text but appear in the appendix and it's not clear that the reference is to an appendix table. Table 2 is not referred to or discussed from the text at all, as far as I can tell.

## Minor comments

- End of section 3: "a models"
- The clause starting with "the mapping between" is missing a verb.

---

> ### Author Response · Authors · 2020-11-24
> **Reply to R1's Summary and Evaluation**
>
> Thanks for your detailed review and for your willingness to re-evaluate your score based on our responses to your feedback. Please find our replies to your comments below and see the updated draft, which reflects your suggested changes.
>
>
> **Replies to “Evaluation” comments:**
>
> 1. **Re: Important question** — thanks!
>
> 2. **Re: emphasis on counterfactually augmented data** — thanks for making this critical point. We are of two minds here. On one hand, we agree that many of our experimental insights speak more broadly to reliance on causal and non-causal features. On the other hand, our toy model gives direct insights into why CAD might work (d-separation of outcome from non-causal features). Additionally, understanding CAD is the primary motivation driving this research. We have attempted to draw an appropriate compromise, keeping the original title but up-emphasizing the broader applicability of our investigation.
>
> 3. **Re: disconnect between toy models and practice** — We agree that considerable work remains to connect theory to practice, both here and in ML more broadly. Our research philosophy is to put the two in dialogue, both to highlight the applicable insights and to make salient the remaining disconnects, even when they are unsatisfying. We have attempted in our discussion to make some connections clearer (in Section 3, for e.g., why we think the noisy proxy may be a useful abstract model for what happens in practice). Overall, we agree with you, and are glad both that you recognize the disconnect, appreciate our efforts to address it, and also the difficulty of the task.
>
> 4. **Re: Clarity of exposition on experiments** Thanks for this feedback. We have cleaned up the writing and broken up the excessively long paragraph and added additional discussion (Section 4).
>
> 5. **Re: Length of introduction** Per your feedback, we have condensed the introduction.

---

> ### Author Response · Authors · 2020-11-24
> **Reply to R1's "Main Comments"**
>
> 1. **what does salient mean here** — here we mean only that as the noise on the causal features grows, the weights on the non-causal features grow in magnitude. We have clarified this in the draft (Sections 1 and 3).
>
> 2. **Meaning of anti-causal interpretation**  In general, in “causal learning” we think of the features as causing the label, while in “anticausal learning” we think of the label as causing the features. We point to “On Causal and Anticausal Learning” (Schölkopf 2012) for a great exposition on the topic.
>
> Here, we mean that rather than thinking of CAD as an intervention by the human editor on the features (which cause the sentiment label to be/not be applicable), we could think of the sentiment as a cause of the review text. In this interpretation, we imagine that we have intervened on the sentiment and the editor’s role is to simulate the counterfactual review that would flow from the alternative sentiment, holding other latent attributes constant. We have revised the draft to make this clearer.
>
> 3. Noise on x2 and noise on a proxy for x2 would be equivalent (with the proxy just increasing the noise with which x2 is observed). Thus we do not need to explicitly model a proxy on x2. For x1, however, because x1 (post noise in generating x1) causes the label, but we only observe the noisy proxy (differentiating the true cause of y from what we observe), these two sources of noise (in generating x1 vs in measuring x1) are different and must be modeled separately.
>
> 4. **Re: meaning of intervention** In causality, **intervention** corresponds to setting a variable to a value, regardless of what value it otherwise would have taken. This can be thought of as mutilating the graph to delete all inbound arrows to that variable (see Pearl’s definitive textbook “Causality” 2009). In a classic example consider an intervention like exercise. We might observe associations between exercise and health outcomes, but these could potentially be explained by common causes (e.g. genes) that influence both propensity to exercise and various health outcomes. To understand the effect of exercise on health, we must address what would happen if we intervene and force someone to exercise, regardless of whether they naturally would (vs forcing them not to).
>
> 5. **Re: more careful discussion**  We have revised the exposition of our experiments to make these details clearer.
>
> 6. **Re: NLI vs sentiment** The NLI experiments were tricky. Because we subsampled only data where 10 or more tokens were rationales, for many examples, most of the tokens were rationales. Thanks for calling our attention to this. We believe that we can improve these experiments with a better subsampling heuristic—to pick examples with comparable numbers of rationale vs non-rationale tokens. We are actively working on these experiments per your suggestion and expect to have updated results in time for the camera ready version.
>
> 7. **Re: automatically-identified rationales and gradient-based attribution** Per your suggestion, we ran additional experiments based on saliency scores obtained from a RoBERTa model (https://docs.allennlp.org/master/api/interpret/saliency_interpreters/saliency_interpreter/) and added the results to the draft (Figs 2c and 3c; Tables 5 and 8). Per computational resources, these preliminary results only reflect one run. We will update the draft again when all 5 runs are complete to report the average.
>
> 8. **Re: noise to non-rationales doesn’t always result in better OOD performance** We note that the real world is complicated, and distribution shifts come in many forms. While relying on non-causal features is clearly bad in the worst case, there may also be scenarios where non-causal predictors remain useful out of domain. While our results are suggestive and lend support to our framing of the problem, we don’t want to misrepresent them as providing a complete account for out of distribution generalization. We discuss this in our updated draft (Sections 3 and 4)
>
> 9. **Re: style transfer experiments seem disconnected from the main theme of the paper** The primary mission of our paper is to better understand CAD, both why it works and what benefits it gives out of domain. One key question is whether we really need to go through the process of collecting CAD (or human-annotated rationales) at all or if automated methods for generating “counterfactuals” might perform similarly. These experiments provide evidence that these automated processes that use the original data alone are insufficient to provide the benefits of CAD. We agree that the original exposition did not make these points sufficiently clear and have improved the exposition to better integrate this portion of our contribution. Thanks for the critical feedback.
>
> 10. **Re: Layout** We are working to improve the layout to make figures and tables more readily accessible. Thanks for the feedback!
>
> Thanks also for catching the typos! We have fixed these issues in the draft.

---

### Author Response · Authors · 2020-11-24
**General reply to reviewers**

We would like to thank all four reviewers for taking the time to provide thoughtful reviews and constructive feedback. We are encouraged to see that 3 reviewers advocate for acceptance and that two appear to champion the paper. Specifically, we were glad that the reviewers thought that our paper addresses an important question (R1, R2, R3, R4), found our toy example to be useful/insightful (R1, R2, R4), our experiments comprehensive (R1, R2, R4), our exposition clear (R4), and that this paper opens a new line of research (R2).

We are also grateful to the reviewers for a number of constructive suggestions. Inspired by their feedback, we ran several additional experiments and have improved the exposition:

Per R1’s suggestion, we extended our comparisons to automatically-identified rationales by **adding experiments using a gradient-based feature attribution method**.

Per R1 and R3’s suggestion, we took some steps to address the conceptual gap between our stylized models and real-world applications of CAD by **adding more critical discussion**.

Per R1’s suggestions, **we trimmed the introduction and extended the discussion of the anti-causal setting**.

We reply to each reviewer’s individual concerns in greater depth in the respective threads.

---

### Decision · Program_Chairs · 2021-01-07
**Final Decision**

**Decision:**

Accept (Poster)

**Comment:**

This paper presents a theoretical characterization of the impact of noise in causal and non-causal features on model generalization, through the lens of counterfactual data augmentations with toy data and models, and demonstrates that the predictions of this characterization bear out in several experiments on language counterfactually-augmented language data with substantial models.

Pros:
- Spurious features and their relationship out-of-domain generalization are a practically issue in modern applied ML, and this work helps to coalesce our understanding of this area.
- Fairly extensive experimental work.

Cons:
- Reviewers didn't find the connection between the theoretical analysis, which focused on a simplified setting, and the experimental work, to be especially clear. In particular, reviewers worried that the predictions that were tested experimentally were fairly intuitive ones that could reasonably be derived from a number of starting assumptions, so it's not clear that they offer strong support for the specific account given here.
- Reviewers found the presentation, especially of the empirical work, confusing.

Overall, this paper makes a legitimate and sound contribution to an important research area. That contribution is small, and somewhat easy to misinterpret, but after some discussion, reviewers agreed that the paper should still be a worthwhile net positive for the field.